# Structure of the HIV immature lattice allows for essential lattice remodeling within budded virions

Sikao Guo[1], Ipsita Saha[2], Saveez Saffarian[3,4,5], Margaret E Johnson[1]*

[1]TC Jenkins Department of Biophysics, Johns Hopkins University, Baltimore, United States; [2]Laboratory of Cell and Developmental Signaling, Center for Cancer Research, National Cancer Institute, National Institutes of Health, Frederick, United States; [3]Center for Cell and Genome Science, University of Utah, Salt Lake City, United States; [4]Department of Physics and Astronomy, University of Utah, Salt Lake City, United States; [5]School of Biological Sciences, University of Utah, Salt Lake City, United States

*For correspondence:
margaret.johnson@jhu.edu

Competing interest: The authors declare that no competing interests exist.

**Abstract** For HIV virions to become infectious, the immature lattice of Gag polyproteins attached to the virion membrane must be cleaved. Cleavage cannot initiate without the protease formed by the homo-dimerization of domains linked to Gag. However, only 5% of the Gag polyproteins, termed Gag-Pol, carry this protease domain, and they are embedded within the structured lattice. The mechanism of Gag-Pol dimerization is unknown. Here, we use spatial stochastic computer simulations of the immature Gag lattice as derived from experimental structures, showing that dynamics of the lattice on the membrane is unavoidable due to the missing 1/3 of the spherical protein coat. These dynamics allow for Gag-Pol molecules carrying the protease domains to detach and reattach at new places within the lattice. Surprisingly, dimerization timescales of minutes or less are achievable for realistic binding energies and rates despite retaining most of the large-scale lattice structure. We derive a formula allowing extrapolation of timescales as a function of interaction free energy and binding rate, thus predicting how additional stabilization of the lattice would impact dimerization times. We further show that during assembly, dimerization of Gag-Pol is highly likely and therefore must be actively suppressed to prevent early activation. By direct comparison to recent biochemical measurements within budded virions, we find that only moderately stable hexamer contacts ($-12k_BT<\Delta G<-8k_BT$) retain both the dynamics and lattice structures that are consistent with experiment. These dynamics are likely essential for proper maturation, and our models quantify and predict lattice dynamics and protease dimerization timescales that define a key step in understanding formation of infectious viruses.

## Editor's evaluation

This fundamental work substantially advances our understanding of the maturation of retroviruses, a key step in understanding the formation of infectious viruses. The evidence supporting the conclusions is compelling, with rigorous computational simulations. The work will be of broad interest to the community of virologists worldwide.

## Introduction

A key step in the lifecycle of retroviruses such as HIV-1 is the formation of new virions that assemble and bud out of the plasma membrane (*Freed, 2015*; *Freed and Mouland, 2006*). These new virions

are initially in an immature state, characterized by a lattice of proteins attached to the inner leaflet of the viral membrane (*Ono et al., 2004*; *Saad et al., 2006*; *Qu et al., 2021*). This immature lattice is composed of Gag, Gag-Pol, and genomic RNA (gRNA), with some accessory proteins known to be included for the HIV-1 virion (*Freed, 2015*). The Gag polyprotein is common to all retroviruses and makes up most of the observed lattice underlying the virion membrane. Within the lattice, 95% of the monomers are Gag (which has six domains), and 5% are Gag-Pol, which has the six-domain Gag followed by protease, reverse transcriptase, and integrase domains embedded within the same polyprotein chain (*Freed, 2015*). The structure of the immature lattice has been partially resolved using sub-tomogram averaging cryotomography (*Schur et al., 2015*), revealing Gag monomers that form hexameric rings assembled into a higher-order assembly via additional dimerization contacts. For maturation and infectivity of HIV virions (*Göttlinger et al., 1989*; *Swanstrom and Wills, 1997*), the Gag proteins within the immature lattice must be cleaved by the protease formed from a dimer of Gag-Pol. Importantly, the lattice covers only 1/3 to 2/3 of the available space on the membrane (*Wright et al., 2007*; *Briggs et al., 2009*). The incompleteness of the lattice results in a periphery of Gag monomers with unfulfilled intermolecular contacts. Recent work showed that these peripheral proteins provide more accessible targets for proteases (*Tan et al., 2021*). Here, we address a distinct question on an earlier step in maturation: does the incompleteness of the lattice allow for dynamic rearrangements that ensure that protease domains embedded within the lattice can find one another to dimerize?

Homo-dimerization of the protease domain is necessary for its initial activation (*Konvalinka et al., 2015*), with recent cryoEM work demonstrating the dimer can form while attached to Gag-Pols (*Harrison et al., 2022*). Once activated, the protease triggers a cascade of cleavage reactions, starting with its own (*Tang et al., 2008*; *Louis et al., 1999*; *Pettit et al., 2004*). After the Gag monomers have been cleaved, the newly released domains assemble within the virion cavity to form the HIV mature capsid (*Konvalinka et al., 2015*; *Lee et al., 2012*). While the HIV protease has thus been studied extensively (*Lee et al., 2012*), the mechanism of the initial steps leading to activation has not been established. Recent measurements indicate that protease activation can occur within ~100 s following assembly of the lattice (*Qian et al., 2022*; *Hanne et al., 2016*). Here, we use computer simulations of assembled Gag lattices with varying energies and kinetic rates of binding interactions to test how lattice structure and stability can support dimerization of the Gag-Pols at this timescale. Our simulations track the spatio-temporal dynamics of coarse structural models of the Gag/Gag-Pol monomers (*Figure 1*) as they diffuse and react with one another in time (stochastic, particle-based reaction-diffusion) (*Varga et al., 2020*). We determine which mechanisms of inhibited activation, large-scale lattice remodeling, or dissociation and rebinding of Gag-Pol molecules promote dimerization as an essential step in understanding viral maturation. We note that once dimerization occurs, the protease becomes activated and can begin cleavage, but we do not address these latter steps here.

Stability of the immature lattice seems to be balanced to promote assembly but also allow for efficient proteolysis and maturation (*Mallery, 2021*); thus mutations and inhibitions that shift the immature lattice stability alter the infectivity of the virus (*Mallery, 2021*). Maturation inhibitors that bind to the immature Gag lattice are thought to stabilize the lattice, preventing cleavage and maturation, which results in loss of viral infectivity (*Keller et al., 2011*; *Kleinpeter and Freed, 2020*). Mutations that impede binding of the immature Gag hexamers to inositol hexakisphosphate (IP6) destabilize the lattice, again affecting infectivity (*Mallery et al., 2019*). The strength of the hexameric contacts is not known, as it is sensitive to co-factors like IP6 and RNA both in vitro (*Kucharska et al., 2020*) and in vivo (*Mallery, 2021*; *Dick et al., 2018*; *Muriaux et al., 2001*). The lattice is linked to the membrane via lipid binding and myristolation (*Ono et al., 2004*; *Saad et al., 2006*), and thus the increased concentration on the budded membrane will drive distinct dynamics and stability than those expected in a 3D volume due to dimensional reduction (*Yogurtcu and Johnson, 2018*; *Guo et al., 2022*). Identifying regimes of binding stabilities and rates that can support assembly and simultaneously support dynamics or remodeling of the immature lattice is thus important for understanding the requirements for forming infectious virions.

With our simulations, we are then prepared to test distinct mechanisms of protease dimerization possible within the immature lattice. Two primary dynamic mechanisms are possible: (1) large-scale remodeling of the lattice could bring together two fragments that contain protease monomers and (2) protease monomers could unbind and reattach at new lattice sites to promote dimerization. Thus, with

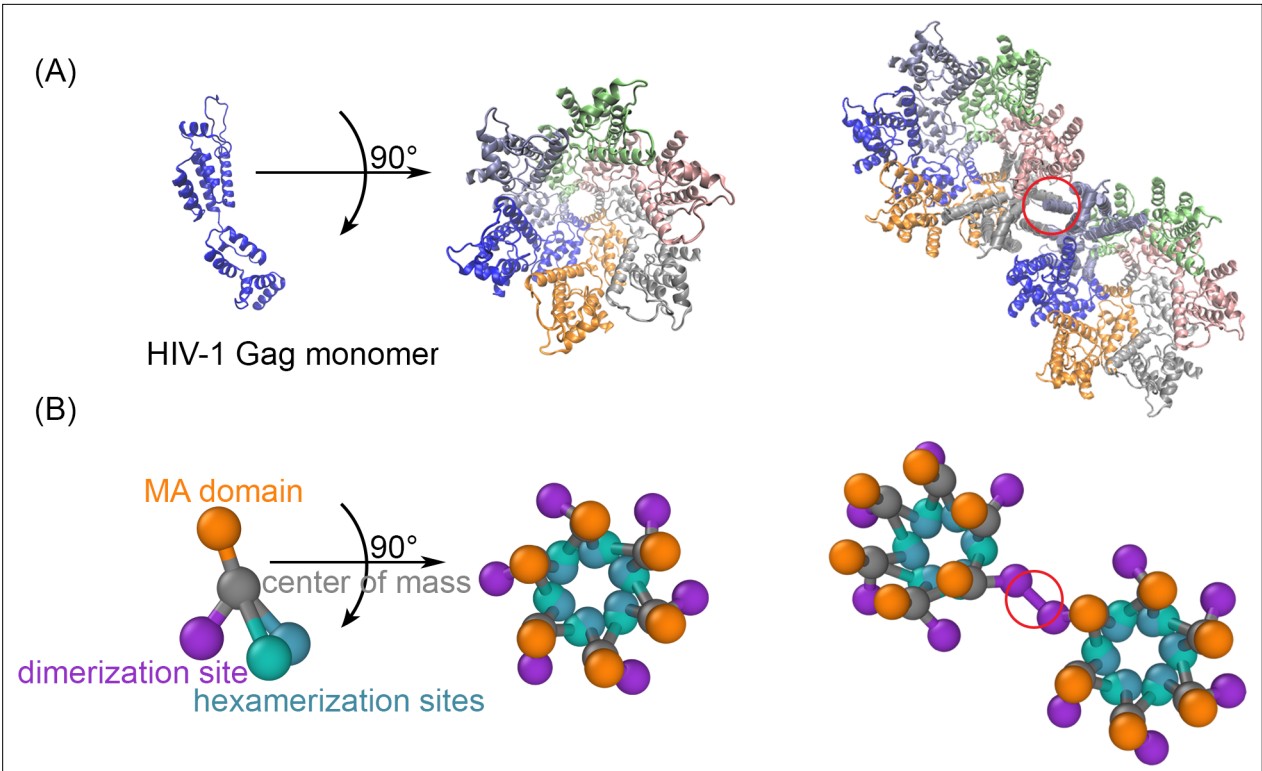

**Figure 1.** The structure-resolved reaction-diffusion model for Gag assembly on spherical membranes. (**A**) The Gag monomers from the cryoET structure of the immature lattice (*Schur et al., 2016*) taken from 5L93.pdb is shown on its own, as part of a single hexamer (center), and with a dimerization interface in the red circle that brings together two hexamers (right). (**B**) Our coarse-grained model is derived from this structure to place interfaces on each monomer at the position where they bind. The reaction network contains three types of interactions. The MA domain (orange) binds to the membrane. The position of the MA site is not in the cryoET structure, and we position it to place each monomer normal to the surface. The distance of the MA site from the center of mass is set to 2 nm. The hexamerization sites (green and blue) mediate the front-to-back binding between monomers to form a cycle. The dimerization site (purple) forms a homo-dimer between two Gag monomers, as illustrated on the right. The reactive sites are point particles that exclude volume only with their reactive partners at the distances shown. Thus, the hexamer-hexamer binding radius is 0.42 nm, whereas the longer dimer-dimer binding radius is 2.21 nm. Positions and orientations are defined in Source Data. The experimental lattice has an intrinsic curvature, and our model recapitulates this to assemble a sphere. The binding kinetics between the interaction types for multiple rates was validated against theory (*Figure 1—figure supplements 1 and 2*), and we verified that the lipid binding site model did not significantly impact the dynamics of the lattice (*Figure 1—figure supplement 3*). The positioning of the Gag interfaces in this model of the immature lattice are distinct from a model that would assemble the mature lattice (*Figure 1—figure supplement 4*).

The online version of this article includes the following source data and figure supplement(s) for figure 1:

**Source data 1.** Model Coordinates.

**Figure supplement 1.** Kinetics of dimer formation between Gag monomers is consistent with theory.

**Figure supplement 2.** NERDSS simulations of purely hexamer assembly in 2D are validated against theory.

**Figure supplement 3.** Comparison of the dynamics of remodeling simulations using the implicit lipid model and explicit lipids.

**Figure supplement 4.** Comparison of the coarse-grained model of Gag monomer from the immature and mature lattice.

Gag-Pols incorporated into the lattice at distinct spatial locations, there must therefore be dynamic remodeling of monomers or larger patches within the lattice. Recent experiments show clear evidence of lattice mobility in virus-like particles (VLPs) (*Saha and Saffarian, 2020*), which are produced by cells expressing only HIV Gag proteins and assemble a Gag lattice very similar to the immature HIV virions (*Gheysen et al., 1989*). Measurements using time-resolved super-resolution imaging and biochemical cross-linking experiments indicate that Gag lattices that cannot undergo maturation nonetheless exhibit large-scale motion and binding events between individual Gag monomers (*Saha and Saffarian, 2020*). Furthermore, structural analysis of the immature lattice indicates that the edges of the ~2/3 complete lattice contain Gag monomers that are attached with fewer links (*Tan et al., 2021*). These 'dangling' proteins would be able to more freely detach and reattach at distinct sites.

From such cryoET structures (*Tan et al., 2021*), however, it is not possible to measure the dynamics of the lattice and Gag monomers. We note a third mechanism allows the proteases to dimerize during assembly, so they are already adjacent. Experimental evidence indicates they would have to remain as an autoinhibited dimer until after budding occurred, to ensure that the full-length Gag is assembled into budded virions (*Lee et al., 2012*). Autoinhibition would prevent early activation of proteases that is known to significantly limit particle formation (*Kräusslich, 1991*), and cause assembly defects (*Ott et al., 2009*). While we cannot test molecular mechanisms of autoinhibition with our model, we can quantify the likelihood of protease dimerization during assembly.

Previous modeling work studying the HIV-1 immature lattice has captured similar structural features to our work but has not interrogated the membrane bound lattice dynamics and their implications for protease dimerization. Coarse-grained molecular-scale models of the immature Gag lattice established interaction strengths between Gag domains that are necessary to maintain a hexagonal lattice ordering, as well as changes in structure following mutation (*Ayton and Voth, 2010*). Molecular dynamics simulations of incomplete hexamers along the immature lattice gap-edge demonstrated conformational changes in Gag monomers that indicate lower stability and likely targets for protease cleavage (*Tan et al., 2021*). Coarse-grained simulations of lattice assembly in solution (*Pak et al., 2022*) and on membranes *Pak et al., 2017* have identified the importance of co-factors, including the membrane, RNA, and IP6 in stabilizing hexamer formation and growth. Similar to these molecular dynamics simulations, our reaction-diffusion simulations also track the coarse-grained coordinates of each Gag monomer in space and time. In contrast, our model is parameterized not by empirical energy functions describing how each site in the model attracts/repels other sites, but instead by rates that control the probability of binding upon diffusive collisions (*Varga et al., 2020*). With this reaction-diffusion approach, we have access to longer timescales despite the large system size (~2500 monomers), and precise control over the association kinetics and free energies, which are directly input as parameters to our model. We can thus quantify the dynamics and kinetics of the assembled lattice over several seconds for multiple model strengths and rates.

In this work, we initialize Gag monomers into their immature lattices on the membrane, as they would be structured after budding from the host cell but prior to maturation (*Tan et al., 2021*). We use reaction-diffusion simulations to both assemble these immature lattices and characterize the timescales of remodeling and Gag dynamics within the incomplete lattices. We validate that our structured lattices conform to those observed in cryoET through a quantitative analysis, and we verify that the specified free energies and rates of association between our Gag monomers are validated in simpler models. We first characterize the likelihood of the Gag-Pol monomers to dimerize during the assembly process. We find that although they represent only 5% of the monomers that assemble into the lattice, the stochastic assembly will ensure that at least a pair of them are adjacent within the lattice, even if they do not engage in a specific interaction. We next show that, if, on the other hand, the molecules are distant from one another, they would need to detach, diffuse, and reattach stochastically at the site of another Gag-Pol molecule. By modulating the kinetics and energetics of Gag-Gag contacts, we quantify how the overall time for dimerization depends on unbinding, and rebinding, with the 2D diffusion contributing negligibly to the overall time. Lastly, we show how the mobility of the lattice causes binding events that are consistent with biochemical measurements (*Saha et al., 2021*), and decorrelation of the lattice that is qualitatively consistent with recent microscopy measurements on immature Gag lattices (*Saha and Saffarian, 2020*). Our results show that the stochastic dimerization of two Gag-Pol molecules would need to be actively suppressed or inhibited to effectively prevent early activation, and that otherwise, even stable lattices can support Gag-Pol dimerization events due to dynamic remodeling.

## Results

### Assembled lattices on the membrane are structurally similar to those present in cryoET

Our model captures coarse structure of the Gag and Gag-Pol monomers as derived from a recent cryoET structure (*Schur et al., 2016*) of the immature lattice (*Figure 1A*) (Methods). The Gag-Pol is structurally identical to the Gag but represents 5% of the total monomer population to track protease locations within the lattice. We were able to assemble a variety of spherical Gag lattices that grew

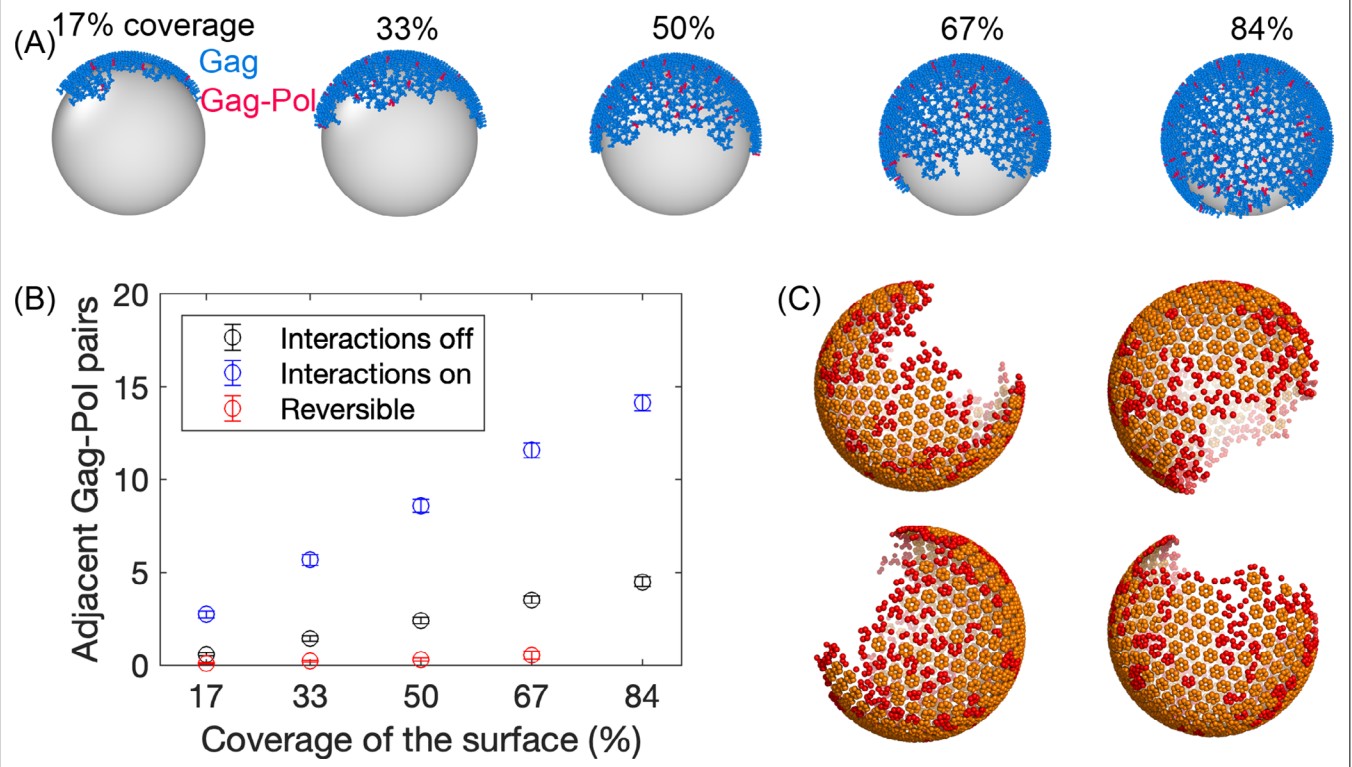

**Figure 2.** Initial Gag immature lattices within the membrane are assembled via simulation. (**A**) The starting Gag immature lattices are assembled from NERDSS simulations with irreversible binding; ~5% of the monomers are Gag-Pols shown in red. We note that the silver spheres are shown here only to improve visualization of just one side of the lattice; Gag proteins are attached to the *inner* surface of the budded spherical membrane, consistent with experiment. (**B**) The number of adjacent pairs of Gag-Pol in the initial immature lattice increases with more surface coverage. Normally, we set all parameters for Gag and Gag-Pol to be identical (blue circles). During assembly, we tested turning off any explicit Gag-Pol to Gag-Pol interactions, rendering them unfavorable (black circles), but they can still end up adjacent to one another. However, this is sensitive to the assembly conditions—when monomers can unbind during assembly, they can correct these unfavorable interactions and reduce the Gag-Pol to Gag-Pol pairs further (red circles). (**C**) Formation of the lattice produces structures that are similar to cryoET, with a single large continent and a large vacancy, as well as several defects or incomplete hexamers throughout the large lattice, which are shown in red in these four independent assemblies. An incomplete hexamer in the simulated lattice is quantified as a sub-structure with 2–5 monomers present in the ring. The size distribution of these defect regions is also found to be similar to the cryoET results (*Figure 2—figure supplement 1*).

The online version of this article includes the following figure supplement(s) for figure 2:

**Figure supplement 1.** Quantitative analysis of the defects in the structures of our model and cryoET.

from monomers to a single sphere with our targeted coverage of the membrane surface using our stochastic reaction-diffusion simulations (*Varga et al., 2020*; *Figure 2*) (Methods). The lattices in *Figure 2* are a single connected continent, with imperfect edges, a large gap on the surface (~1/3), and regions with defects present in the tri-hexagonal lattice (*Figure 2*). Our lattice topologies are in very good agreement with the structures determined by cryoET, which also shows a single continent and a large gap, such that the spherical lattice is truncated (*Tan et al., 2021*). We quantified the fraction of hexamers in our lattices that are incomplete, finding 36–40% have fewer than 6 monomers when binding events during assembly are irreversible, or 30–32% when we allow unbinding during assembly (see Methods). This is in excellent agreement with the 34%±4% we calculated from the cryoET datasets (*Tan et al., 2021*). We also observe a similar distribution in the sizes of the regions containing incomplete hexamers, with most regions being localized and small, but with a few larger strands or 'scars' (*Figure 2—figure supplement 1*). Along the incomplete edge, we count a larger fraction of the free binding sites are hexamer sites, in agreement with experiment (*Tan et al., 2021*), although we acknowledge our simulations do not exclude free dimer sites (which are not observed in the cryoET) given the assembly parameters (dimer and hexamer rates are equally fast).

It is illuminating that the structures of our lattices share features with the experimental lattices, given that our assembly simulations (see Methods) do not directly mimic the physiologic process of Gag assembling in the cytoplasm, at the plasma membrane, with RNA (*Tritel and Resh, 2000*). To promote the nucleation and growth of only a single lattice (rather than nucleating multiple lattice structures), we combined fast Gag-Gag binding ($6 \times 10^6$ M$^{-1}$s$^{-1}$) with a slow titration of Gag monomers into the volume. The slow titration does mimic the role of co-factors, however, in that Gag does not assemble without being effectively 'turned on' by co-factors like RNA (*Qian et al., 2023*). The similarity of our structures to experiment suggests that our assembled model is constrained to incorporate topological defects at a similar frequency to the biological proteins. Interestingly, while these lattices must have defects because a sphere cannot be perfectly tiled by a hexagonal lattice, the number of non-hexamers or imperfect contacts within them is significantly higher than the number required by Euler's theorem, which is only 6 for a spherical lattice with a hole in it (*Negri et al., 2015*). We speculate that during assembly, the lattice is not undergoing a significant amount of remodeling and annealing to correct these defects. This would be consistent with a fast and more irreversible nucleation and growth, and indeed we see fewer defects (~31% vs 38%) when we allow for unbinding during assembly vs irreversible binding. The biological lattices seem to be 'good enough' despite the possibility of more perfect lattice arrangements, and the lower stability of these more defective lattices should facilitate the remodeling necessary for maturation.

## A pair of Gag-Pol monomers are highly likely to stochastically assemble adjacent to one another within the immature lattice

Although only 5% of the Gag monomers in our simulation are tagged as Gag-Pol (~125 out of ~2625 simulated proteins), we find it is extremely unlikely that a lattice will be assembled without a pair of them already adjacent (*Figure 2B*). This is due to the stochastic nature of the assembly and the fact that each monomer has 3 adjacent monomers, two via its hexamer interfaces and one via its dimer interface. However, we can reduce the number of Gag-Pol to Gag-Pol pairs if we turn off any specific interaction between them by setting their binding rates to 0. Even making this interaction thus highly unfavorable relative to a Gag to Gag or Gag to Gag-Pol interaction, we still find pairs of them adjacent, as they can be brought into proximity via their specific interactions with the Gag monomers (*Figure 2B*). The number of pairs given the unfavorable interaction is also dependent on the assembly conditions; when we allow for unbinding between Gag contacts this allows for annealing and correction of such unfavorable contacts during assembly, and the Gag-Pol to Gag-Pol pairs are largely eliminated. Overall, these results indicate that to prevent early activation of the proteases, one cannot just rely on the lower frequency of Gag-Pol to Gag-Pol interaction, as the lattice is simply too densely packed. Instead, these Gag-Pol dimers would have to be actively inhibited from initiating protease activity by either having a highly unfavorable affinity for one another or otherwise forming dimers that are enzymatically inhibited, as any activation preceding budding can leak proteases back to the cytoplasm (*Bendjennat and Saffarian, 2016*), and is known to reduce infectivity (*Kräusslich, 1991*). Regardless of how the activation is prevented, inhibition would have to be released following budding, and this mechanism is not known. We assume below that the Gag-Pol to Gag-Pol dimers following budding can now interact favorably, identically as Gag to Gag, since we know that activation must ultimately occur.

## The Gag lattice disassembles with the weaker hexamer contacts of $-5.62k_{\mathrm{B}}T$

We perform all our simulations from the same starting structures, but with a range of hexamer strengths of $-5.62$ to $-11.62k_{\mathrm{B}}T$, and a slower (0.015 µM$^{-1}$s$^{-1}$), medium (0.15), and faster (1.5) rate of binding for each $\Delta G_{\mathrm{hex}}$. For the weakest hexamer contacts of $-5.62k_{\mathrm{B}}T$, we find that the lattice is not able to retain its single continental structure, and instead fragments into a distribution of much smaller lattices (Video 2). Given a fixed $\Delta G_{\mathrm{hex}}$ we speed up the on- and off-rates and as expected, we see more rapid disintegration of the lattice structure. As we stabilize the lattice by increasing $\Delta G_{\mathrm{hex}}$, we still see departure from the single continental structure due to unbinding of monomers and small complexes from the lattice edge (*Figure 3*). Hence, we see the emergence of a bimodal distribution of lattices, with a peak at the monomer/small oligomer end, and another peak containing the majority of the lattice in one large continent. The size of the large continent remains largest with increasing $\Delta G_{\mathrm{hex}}$

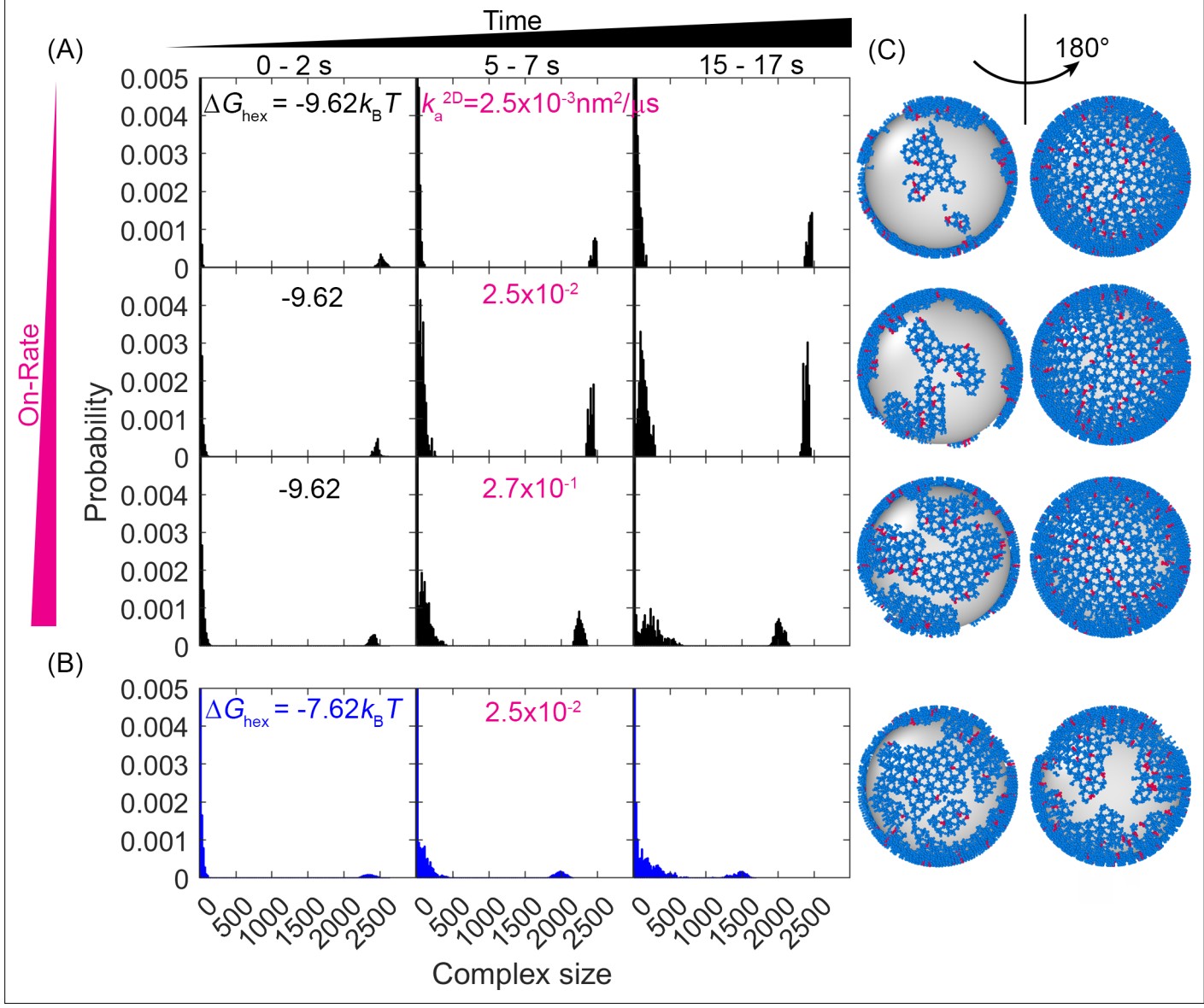

**Figure 3.** Evolution of the lattice size distribution at different reaction rates and hexamer interaction strengths. (**A**) Along the *x*-axis are the numbers of monomers found in each lattice, which is largely bimodal for all systems: a population of small oligomers and one giant connected component. As time progresses (from left to right columns), the initial structure which was one giant connected component continues to fragment somewhat, indicating that the starting structure was not at equilibrium. As the on- and off-rates increase (from top to bottom) with a fixed $\Delta G_\text{hex} = -9.62k_\text{B}T$, the largest component shrinks, as shown by the peak denoting the large giant component shifting to the left, and the peak denoting the small oligomers shifting to the right. (**B**) For a weaker hexamer free energy shown in the blue data ($\Delta G_\text{hex} = -7.62k_\text{B}T$), the lattice is breaking apart more rapidly and moving toward a more uniform distribution of lattice patch sizes as both peaks shift to the center. Note that we cut off the *y*-axis at 0.005 to make the peak at ~2500 visible. The bars at small sizes extend up to ~0.05. (**C**) Representative structures at the later times (*t*=17 s) for each case, illustrating the increased fragmentation as the rates accelerate, or as the hexamer contacts destabilize (lowest row). We quantify the corresponding diffusivity of the structures in *Figure 3—figure supplement 1*. We show how changes to $\Delta G_\text{strain}$ have a minimal impact on the structural dynamics in *Figure 3—figure supplement 2*.

The online version of this article includes the following figure supplement(s) for figure 3:

**Figure supplement 1.** Distribution of the diffusion constant of each molecule.

**Figure supplement 2.** Comparison of complex size distribution over the first 0–1 s of simulations with two values of $\Delta G_\text{strain}$ .

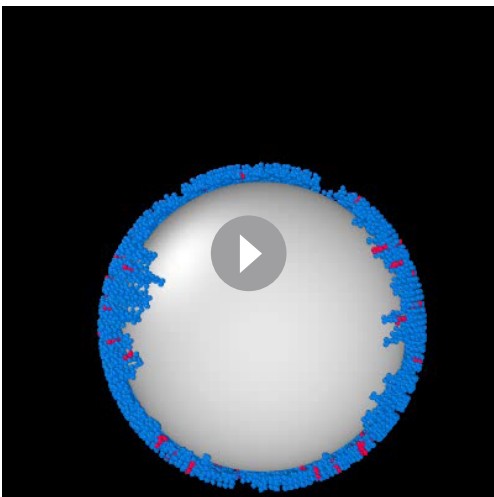

**Video 1.** Lattice dynamics for moderately stable dynamics that are consistent with structural and biochemical experiments. $\Delta G_{\text{hex}} = -9.62 k_{\text{B}}T$. $k_{\text{a}}^{\text{2D}}$ ($\text{nm}^2/\mu s$)=$2.5 \times 10^{-2}$; gap between each frame: 10 ms; overall time length: 20 s.
https://elifesciences.org/articles/84881/figures#video1

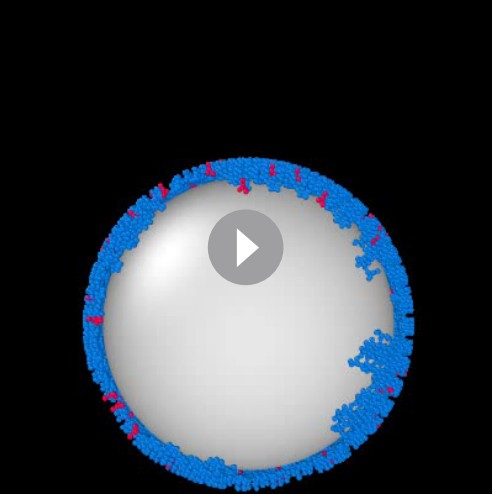

**Video 3.** Lattice dynamics for a more highly stable lattice with slower binding kinetics. $\Delta G_{\text{hex}} = -11.62 k_{\text{B}}T$. $k_{\text{a}}^{\text{2D}}$ ($\text{nm}^2/\mu s$)=$2.5 \times 10^{-3}$; gap between each frame: 10 ms; overall time length: 20 s.
https://elifesciences.org/articles/84881/figures#video3

and with a slower rate, over the course of these ~17–20 s simulations (*Figure 3*). Importantly, these dynamics occur in all our simulations and would not be possible if not for the incompleteness of the lattice. Specifically, the Gag contacts are dissociating not from the membrane but from each other, predominantly along the edge, at which point they can then diffuse along the membrane surface (*Video 1*, *Video 2*, *Video 3*). If the lattice were covering 100% of the surface, dissociation events would not allow Gags to diffuse away, and no dynamic remodeling would occur. From the sizes of the lattices present in the simulations, we can also report on the distribution of diffusion constants represented on the surface, as larger lattices diffuse more slowly. For the weaker lattices, the distribution is very broad, spanning 4 orders of magnitude, whereas for the most stable lattice there is primarily one very slowly diffusing timescale, and a separate timescale for the more faster moving oligomers (*Figure 3—figure supplement 1*). Lastly, the lifetimes of hexamers in our lattices are controlled by $\Delta G_{\text{hex}}$, by a $\Delta G_{\text{strain}}$ penalty, and by the extent to which the hexamers are constrained by further dimer contacts. Our $\Delta G_{\text{strain}}$ penalty is small at $2.3 k_{\text{B}}T$ but it does shorten the hexamer lifetimes relative to having 0 strain (Methods). This means that the strain penalty can increase lattice dynamics, but we see that the relaxation dynamics from the initial lattices is much more sensitive to the magnitude of $\Delta G_{\text{hex}}$ (*Figure 3*). The effect becomes negligible for more stable lattices as the remodeling we observe is dominated by Gag subunits on the edge that form incomplete hexamers. Overall, if the value of $\Delta G_{\text{strain}}$ were large ($\sim\Delta G_{\text{hex}}$) it could impact at which value of $\Delta G_{\text{hex}}$ the lattice transitions from a primarily single-connected component to the fragmented lattice we see here at $\Delta G_{\text{hex}}$=–5.62$k_{\text{B}}T$.

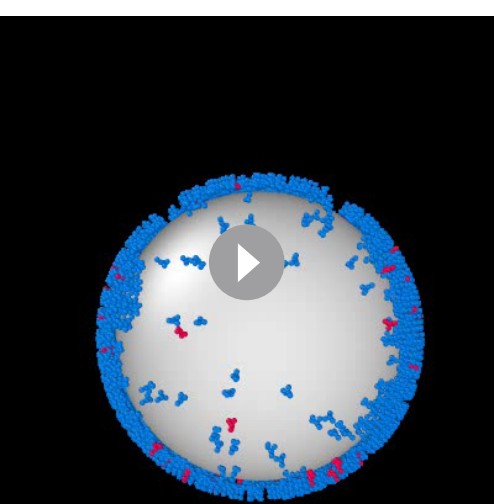

**Video 2.** Lattice dynamics for an unstable lattice. $\Delta G_{\text{hex}} = -5.62 k_{\text{B}}T$. $k_{\text{a}}^{\text{2D}}$ ($\text{nm}^2/\mu s$)=$2.5 \times 10^{-2}$; gap between each frame: 10 ms; overall time length: 4.5 s.
https://elifesciences.org/articles/84881/figures#video2

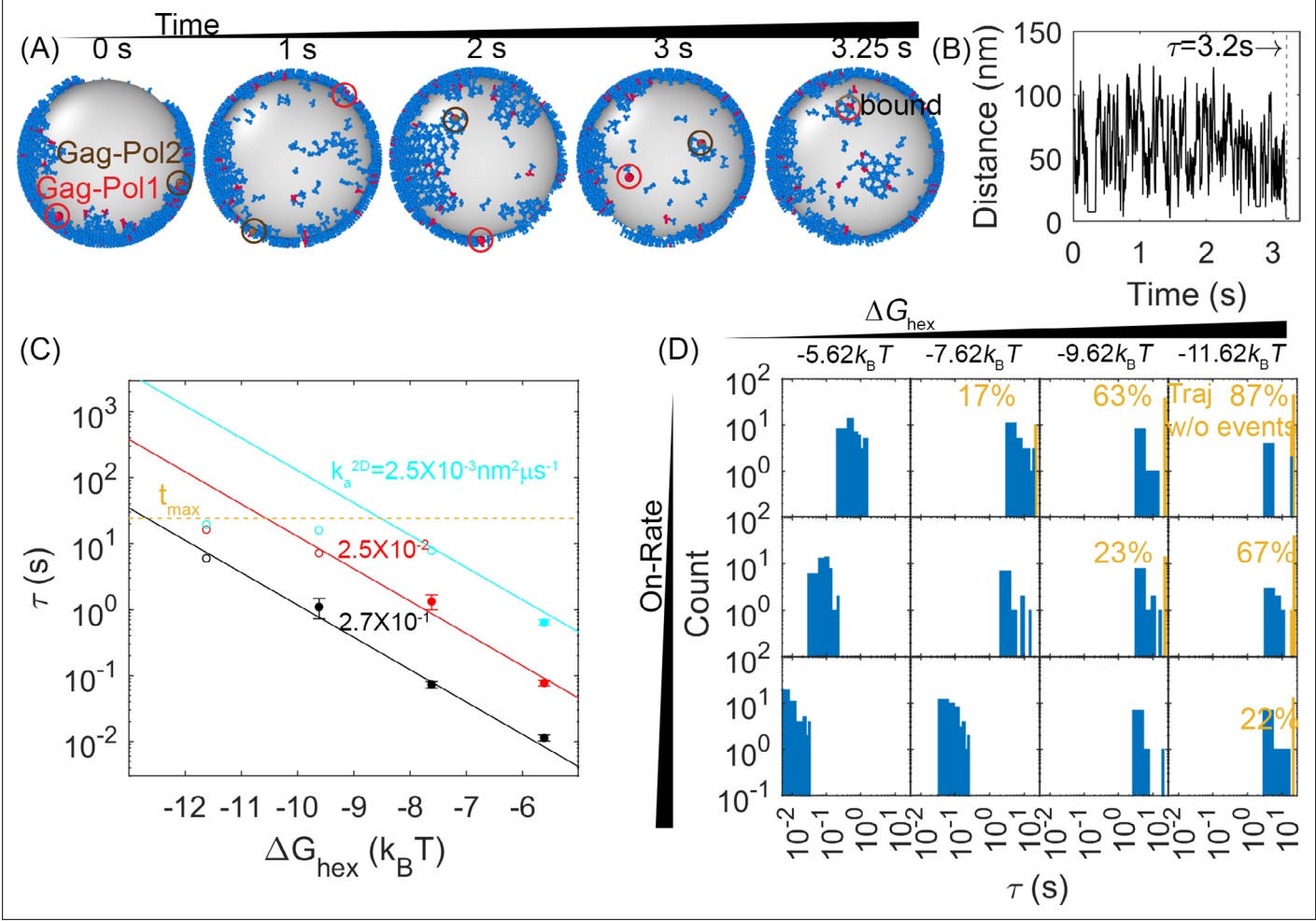

**Figure 4.** First-passage times (FPTs) for a pair of Gag-Pol monomers to search and bind to one another reveal a clear dependence on hexamer rates and free energies. (**A**) An example from a simulation of how two Gag-Pols found each other. Characterization of the edge connectivity in *Figure 4—figure supplement 1*. (**B**) The distance between all Gag-Pol pairs can be monitored in time, with this trace corresponding to the simulation in (**A**). The distance fluctuates and drops to the binding radius $\sigma$ at 3.2 s, after which the two molecules remain bound. (**C**) FPTs of Gag-Pol dimerization at different reaction rates and hexamer free energies $\Delta G_{hex}$. The yellow dashed line indicates the maximal length of the simulation traces. The filled-in circles report the mean FPT (MFPT) for parameter sets where all traces produce a Gag-Pol dimerization event. The open circles report a lower bound on the MFPT, because some of the traces were not long enough to observe a Gag-Pol dimerization event. The solid lines are the fits to the FPTs from using *Equation 1*, using only data points that had at least 75% of the trajectories produce dimerization events. The adjusted $R^2$ measure for the fit is 0.98, which accounts for the small sample size. The leftmost red point and two leftmost cyan points are excluded from the fit because the absence of dimerization events exceeds 25% in these cases. If we fit only points with 100% of trajectories completed, we recover the same power law trends with slightly different parameters. (**D**) The distributions of the FPTs at different reaction rates (each row) and hexamer free energy (each column). The yellow bars and yellow numbers report the percent of traces without any Gag-Pol dimerization event over the time simulated, so they are placed at the end of the simulated time. The FPTs slow as the reaction rates decrease and as the hexamer contacts become more stabilized.

The online version of this article includes the following figure supplement(s) for figure 4:

**Figure supplement 1.** Characterizing bonds at the edge of the lattice.

## First-passage times for protease dimerization events are dependent on the free energy and binding rates of the hexamer contacts

A primary goal is to characterize the path and the timescale by which two Gag-Pol molecules could find one another given the single continental lattice they are embedded in at time 0. From our initial lattices, we showed in *Figure 2* that at least one pair of Gag-Pol monomers are already in contact with one another, so we ignore those pairs to focus instead on spatially separated Gag-Pols. By tracking the separation between all pairs of Gag-Pol monomers (*Figure 4*), we can quantify the first-passage time (FPT), or the time for the first pair to find one another in each simulated stochastic trajectory.

The edge of the incomplete lattice supports multiple detachment events (*Video 1*); of the ~250 Gag monomers along the edge, 10% have only one link to the lattice, which offers the easiest path to disconnect, by breaking only a single bond (*Figure 4—figure supplement 1*).

In *Figure 4*, our results show how the FPT for two Gag-Pols to dimerize with one another is dependent on both unbinding rates and binding rates, as both events are required to bring two Gag-Pol together. We observe two intuitive trends. One is that for a given free energy $\Delta G_{hex}$, a faster association (and thus faster dissociation) rate results in faster dimerization events between the Gag-Pol monomers. The second trend is that as the lattice free energy stabilizes, dimerization events are slowed due to the slower dissociation times, despite having the same on-rates (*Video 3*). These timescales are thus consistent with the dimerization events requiring at least one of the monomers to dissociate from the lattice, and then rebind at a new location containing a Gag-Pol. We report the association rates as their 2D values, because binding is occurring while the proteins are affixed to the 2D membrane surface, as unbinding from the surface is rare (Methods). The corresponding 3D rates are representative of slow to moderately fast rates of protein-protein association ($1.5\times10^4$–$1.5\times10^6$ $M^{-1}s^{-1}$), where they are converted to 2D values via a molecular length-scale $h$=10 nm (Methods). Activation requires explicitly that the 5% of monomers carrying the proteases to be involved. Hence, additional unbinding and rebinding events will occur that are not 'activating' because they involve a Gag monomer without a protease. Gag-Pol molecules can also unbind and rebind multiple times before successfully finding another Gag-Pol.

## MFPTs can be well approximated and predicted as a function of $\Delta G_{hex}$ and binding rate $k_a$

For the models with weaker and/or faster interactions, our simulations were long enough that all trajectories of that model ($N_{traj}$~60) resulted in a Gag-Pol dimerization event. We could thus construct the full FPT time distribution and reliably calculate the mean first passage times, $\tau_{MFPT}$. The MFPTs displayed a remarkably clear functional dependence on both the $\Delta G_{hex}$ and $k_a$ values as shown in *Figure 4*, for models where the sampling was complete. In contrast, for the most stable lattices with the slowest rates, the majority of the trajectories had not yet produced a dimerization event (*Figure 4D*, yellow bars), and thus the simulations do not report on the true MFPT. We therefore proposed a formula to fit the completed MFPT values, first by analogy to an MFPT model for bimolecular association, with an inverse dependence on $k_a$ (see, e.g., *Mishra and Johnson, 2021*). Second, we empirically find that the $\tau_{MFPT}$ also has a power-law dependence on the $K_D$, $\tau_{MFPT} \propto K_D^{\gamma}$, or equivalently, $\tau_{MFPT} \propto \exp\left(\frac{\gamma\Delta G}{k_BT}\right)$. Our phenomenological formula thus had two fit parameters, the power-law exponent $\gamma$ and a constant pre-factor (see Methods). After fitting, we find the approximate relationship:

$$\tau_{MFPT} = 7 \times 10^{-5} \left(k_a^{2D}/\left(4\pi R^2\right)\right)^{-1} \exp(-1.13\Delta G_{hex}/k_BT) \tag{1}$$

where $7\times10^{-5}$ is a dimensionless fit parameter, $R$ is the radius of the sphere, and our convention has $\Delta G_{hex} < 0$. We see excellent agreement between our formula and our data (*Figure 4C*), which allows us to extrapolate the remaining models beyond the maximum simulation time of 20 s.

## For our models, activation events occur in less than a few minutes

Importantly, in the models we have studied, the activation of a dimer can occur in well under a minute up through several minutes (*Figure 4*). For hexamer stabilities of –5.62 and –7.62$k_BT$, all rates support dimerization events at less than 10 s. For the more stable lattices of –9.62 and –11.62$k_BT$, only the medium and fast rates ensure an MFPT that is less than or comparable to (~50 s): an event occurring within 100 s of the start. Using our *Equation 1*, we can determine that for a moderate rate of $2.5\times10^{-2}$ $nm^2/\mu s$, a $\Delta G_{hex}$ more stable than –12$k_BT$ will be slower than 100 s. For the slowest rate of $2.5\times10^{-3}$ $nm^2/\mu s$, anything more stable than –9.4$k_BT$ will be slower than 100 s. Our most stable lattices at the slowest rates take 10 min on average for an activation event. Our results thus quantify and predict how the kinetics and the stability of the lattice must be tuned to allow sufficiently fast dimerization events involving the 5% of Gag-Pol molecules carrying proteases.

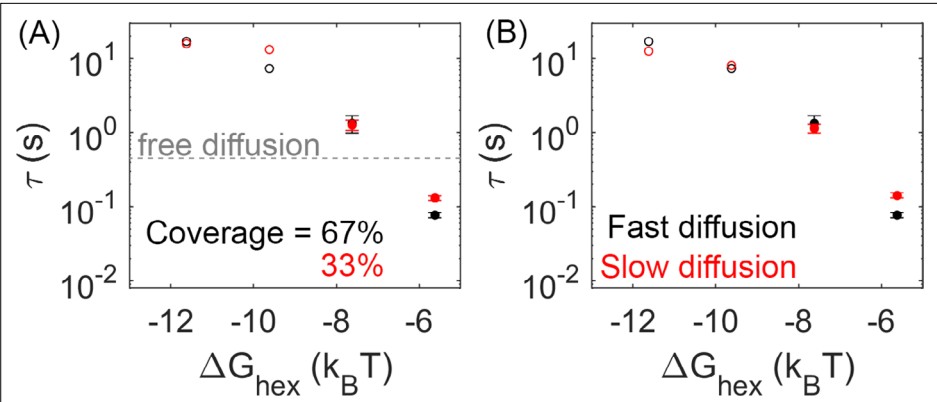

**Figure 5.** Lower lattice coverage or slower diffusion does not dramatically change the mean first-passage time (MFPT). (**A**) First-passage time at different lattice coverages (67% and 33%). Closed circles are cases where Gag-Pol dimerization occurs in 100% of trajectories, while open circles are cases where Gag-Pol dimerization occurs in <100%. The gray line is the first-passage time when two free Gag-Pols diffusing on an empty spherical surface bind to one another. For the weakest lattice, rebinding is actually faster than diffusional encounter times between a dilute pair. (**B**) MFPT at different diffusion constants of the lipid. A monomer of Gag on the membrane diffuses at 0.2 $\mu$m$^2$/s (black data), and diffuses slower as it grows in size consistent with Einstein-Stokes (Methods). We also simulated the system where diffusion of all species was slowed by a factor of 10 (red data). The MFPT is also not sensitive to changes in the dimer strength (*Figure 5—figure supplement 1*).

The online version of this article includes the following figure supplement(s) for figure 5:

**Figure supplement 1.** Effect of dimer interaction strength of mean first-passage time (MFPT) is minimal for these physiologic values.

## Lower lattice coverage does not dramatically change the FPTs

When comparing lattices with 66% coverage vs 33% coverage, we see in some cases a minor slow-down in dimerization times, but the MFPT is overall much less sensitive than it is to the binding rates. With 33% coverage, the edge of the lattice does have a comparable size to the 66% lattice, but the 'bulk' interior is smaller, with more free space required to diffuse to a partner. However, most significantly, the concentration of Gag is smaller, and now with only 66 Gag-Pols (vs 125) present in the lattice, we see in some cases an increase in the time it takes for a pair to find one another (*Figure 5A*).

## FPTs are not sensitive to diffusion or dimerization strengths within a physically relevant range

We tested two strengths for the dimerization free energy of $\Delta G_{\text{dimer}}$ –11.62$k_B T$ and –13.62$k_B T$, comparable to experimentally measured values of dimerization in solution (*Datta et al., 2007*). The MFPTs were overall relatively similar across both values, indicating that the timescales do not show the same sensitivity to $\Delta G_{\text{dimer}}$ as $\Delta G_{\text{hex}}$ over this range of free energies (*Figure 5—figure supplement 1*). This likely emerges because these dimer contacts are typically more stable than the hexamer. Thus, the hexamer unbinding events are more frequent and more likely to directly provoke the first activation events. Further, the hexamer has two binding sites, so more contacts in the lattice, and because hexamers nucleate stable cycles needed for higher order assembly, the frequency of hexamers vs incomplete hexamers are significantly more sensitive to $\Delta G$ than a single dimer bond. This result shows that breaking and formation of the hexamer contacts is important in driving Gag-Pol dimerization events, given that the MFPT shows clear sensitivity to these rates.

We similarly found minimal dependence of the MFPT on the diffusion constant. With a 10-fold slower diffusion constant for all membrane-bound Gag monomers, which effectively slows all lattices down by 10-fold, the MFPTs were not significantly slower (*Figure 5B*). This is not surprising given that the diffusional search along the membrane to find a new partner is not ultimately the rate-limiting step in the association process. The rates we report are intrinsic rates that control binding upon collision, whereas the macroscopic rates one measures through standard biochemistry experiments in the bulk are dependent on both this intrinsic rate and on diffusional times to collision (*Collins and Kimball,*

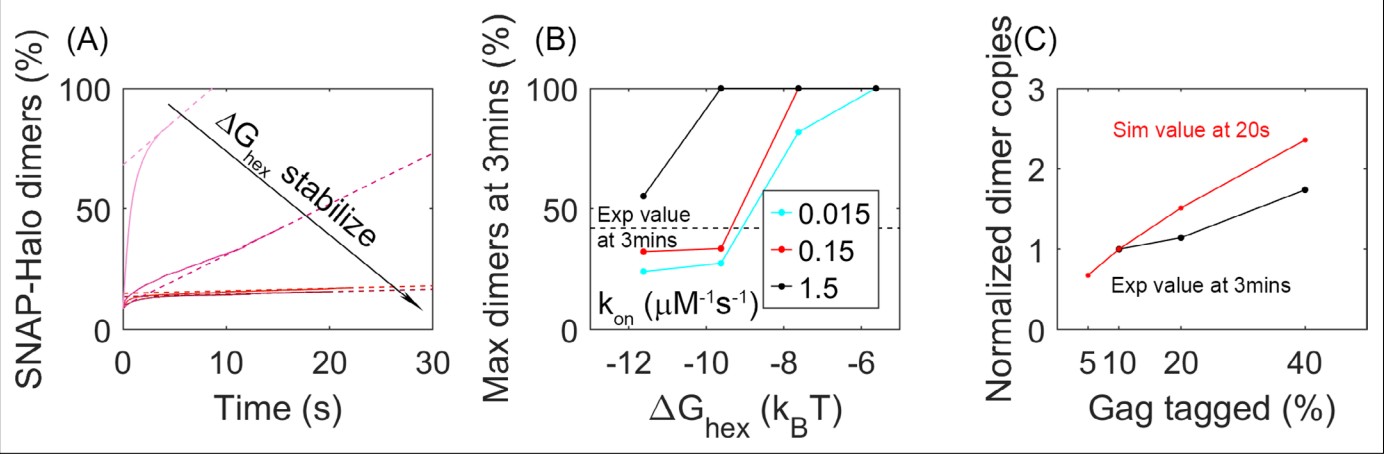

**Figure 6.** Analysis of our simulations mimics experimental biochemical measurements of Gag dimerization as a function of time, agreeing for moderately stable lattices. (**A**) Percent of tagged Gag molecules that have formed a dimer (involving a SNAP-tag and HALO-tag plus linker) as a function of time. Here, 10% of Gag monomers were initially tagged either HALO or SNAP. As the hexamer stability $\Delta G_{hex}$ increases, the dimerization yield dramatically slows. The dashed lines are linear fits of the last 1s of the curves. Results averaged over all 60 traces per parameter set. (**B**) Yield of dimers formed at 3 min estimated via simple linear extrapolation. Our results represent an upper bound. Dashed black line is the experimental measurement of the dimer formation at 3 min given 10% tagged populations. Fast (black), moderate (red), and slower (cyan) rate constants. With the most stable lattices and slowest rates, dimer yield is too low compared to experiment. (**C**) Dimer yield as we increase the population of initially tagged Gag monomers from 5% to 40%. Red is simulated yield at 20 s (to avoid extrapolation assumptions), and black is experimental yield at 3 min. We normalize the yield by the value at tagged Gag = 10%, given the different time points used.

*1949*). Faster intrinsic rates are more diffusion-limited and produce binding that is more sensitive to diffusion (*Yogurtcu and Johnson, 2015*). However, given the small dimensions of the virion, traveling ~70 nm for example (the radius) takes on the order of milliseconds for monomers and small oligomers of Gag. Rough estimates of delay times for a binding event, using $t \sim (k_a^{2D} \ast N_{gagpol}/SA)^{-1}$ indicate that even for the fastest binding, it is on the order of a few milliseconds. The slowest timescale given all the rates is for the stable lattice, where dissociation has a timescale of ~7 s. Altogether, these timescales of individual steps show that the observed MFPTs are not merely controlled by the slowest single events, but by the need for multiple attempts of un- and rebinding to ensure a pair of Gag-Pols find one another. Our results further illustrate how the crowding due to the lattice on the surface can actually accelerate rebinding events compared to a freely diffusing pair when the lattice is unstable (*Figure 5A*), whereas for stronger Gag contacts the lattice will dramatically slow rebinding.

## Biochemical measurements of Gag mobility in VLPs agree with our moderately stable lattices

We find that the dynamics of our simulated lattices agrees with experimental measurements of binding within the lattice for parameters that exclude the most stable, slowest regimes. Experimental measurements within the Gag lattice of budded VLPs tracked the biochemical formation of a Gag dimer involving a population of Gag molecules tagged with a SNAP-tag (10–40%) and the same fraction of Gag molecules tagged with a HALO-tag (*Saha et al., 2021*). A covalently linked dimer was formed through addition of a HAXS8 linker at time 0, with one linker forming an irreversible bridge between a HALO and SNAP protein. Formation of this covalently linked dimer was quantified to reveal an initial rapid formation of dimers, followed by an increasing slower growth that reaches 42% dimer pairs formed for the 10% tagged populations. In our simulations, we thus performed a comparable 'experiment' given our trajectories (see Methods). We tracked the encounter between two populations of our Gag molecules that had been randomly tagged as either 10% HALO or 10% SNAP (*Figure 6*). We similarly found that the majority of the dimers formed rapidly, because they were already adjacent in the lattice when the covalent linker was introduced. The dynamics of the lattice then allowed a slow growth in additional dimers (*Figure 6A*). We calculated the fraction bound over the course of our 20 s simulations and used a simple extrapolation to define an upper bound on the number formed at 3 min (see Methods). For the least stable lattices, the upper bound is close to all

dimers formed, over all three association rates, which is much higher than observed experimentally. For the most stable lattices in contrast, even when our model assumes maximal efficiency of the covalent linker, we extrapolate to an upper bound that is less than 42% dimers formed experimentally (*Figure 6B*). These results thus indicate that these lattices are too stabilized to support the dynamics observed in the Gag VLPs.

We also verified that these simulations are consistent with the trends expected as the population of tagged Gag monomers is increased. Indeed, we find, similar to experiment, that as a larger fraction of Gag monomers have tags, corresponding to a higher concentration of binding partners, we see a larger fraction of dimers being formed (*Figure 6C*).

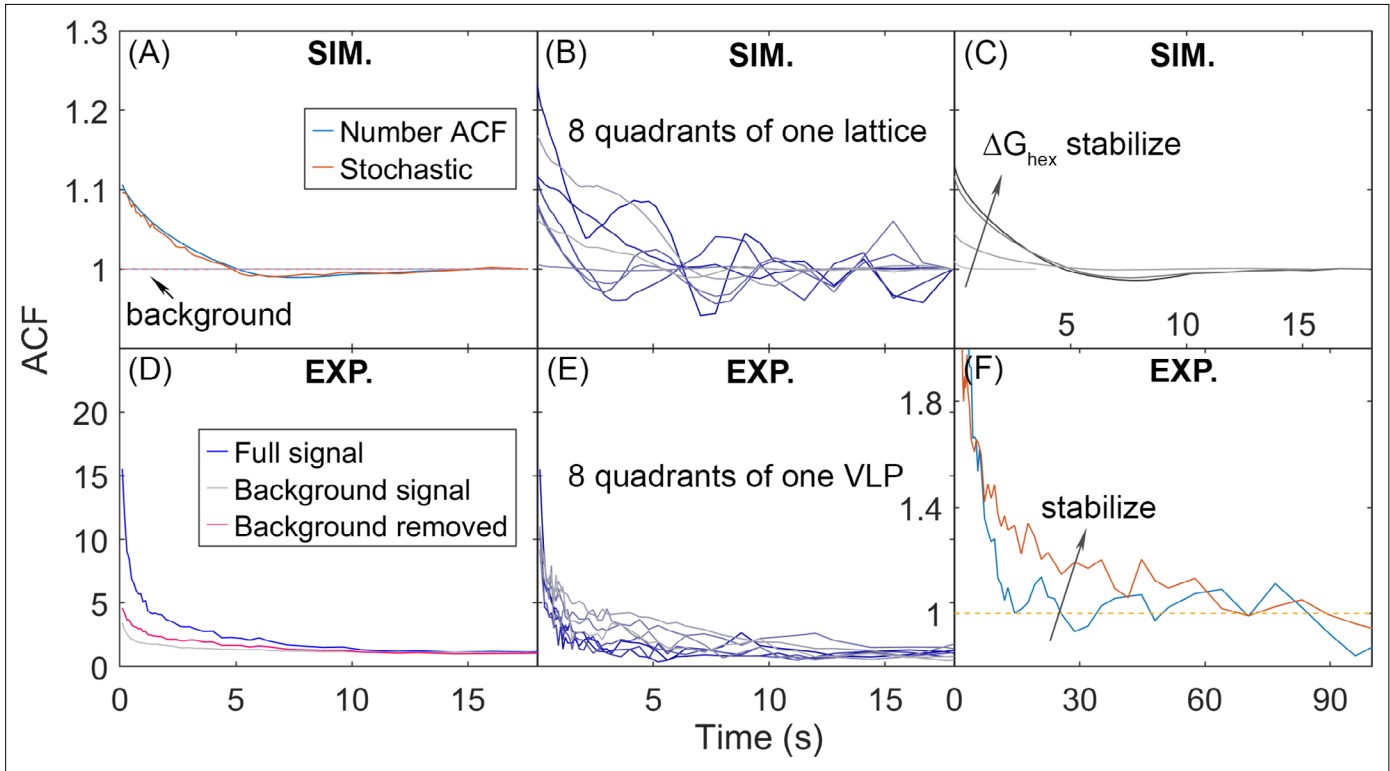

**Figure 7.** Autocorrelation functions (ACFs) of lattice dynamics from simulation and experiment show qualitatively similar trends. (**A**) Number ACF of the simulations calculated directly from the copy numbers of Gag monomers shown in blue line. Averaged over all 8 quadrants over all 60 traces for one parameter set (see Methods). Using the stochastic localization method that mimics experiment shows excellent agreement (orange line). Dashed lines are the background signal, which is 1 as expected (bleaching of the Gag monomers causes limited drops in total copies across 20 s), as the total copy numbers across the membrane surface do not change. We note that the ACF values at our longest delays (i.e. $\tau > \sim 10$ s) are not statistically robust, because of the limited number of frames separated by these timescales. (**B**) ACF of each of the 8 quadrants of one simulated lattice. (**C**) As the lattice is stabilized by increasing $\Delta G_{hex}$, the ACF shows higher amplitude correlations that decay to 1 at longer times, additional trends shown in *Figure 7— figure supplement 1*. (**D**) ACF from stochastic localization experiments on Gag virus-like particles (VLPs). The blue curve is the average signal over all 8 quadrants over 11 VLPs. The gray is the background signal for the ACF of the total copy numbers across the surface, then averaged over all VLPs. The red line is the ACF signal after dividing out the background. (**E**) ACF of 8 quadrants of one experimental VLP. (**F**) The ACF from VLPs that have been stabilized with a fixative (orange curve) show the same trend as the stabilized lattices from simulation. The $y$-axis has been zoomed in to demonstrate the shift. The influence of experimental measurement noise on simulated ACFs is shown in *Figure 7—figure supplement 2*.

The online version of this article includes the following figure supplement(s) for figure 7:

**Figure supplement 1.** Autocorrelation functions (ACFs) at different free energies, reaction rates, diffusion, and surface coverage.

**Figure supplement 2.** Effects of introduced noise on the autocorrelation functions (ACFs) calculated from stochastic localization measurements on simulation trajectories.

## Large-scale and heterogeneous lattice dynamics are visible in autocorrelation functions, and are qualitatively similar to microscopy experiments

We quantify the dynamics of the Gag lattice on fixed viewpoints on the spherical surface using number autocorrelation functions (ACFs), which report on correlations of collective motion that can emerge due to heterogeneity within the lattice (*Figure 7*) (Methods). We expect this heterogeneity due to our lattices all exhibiting a large component and smaller oligomers (*Figure 2*). We find that as the lattice becomes more stable and the bimodal separation of lattice sizes becomes more pronounced, the measured correlations in Gag copy numbers per quadrant increase in amplitude and slow in timescales, and that these dynamics are sensitive to slowing diffusion (*Figure 7—figure supplement 1*). All the ACFs will eventually asymptote to 1 at long delay times as the copy numbers become independent (*Figure 7A*).

In *Figure 7*, we show how the ACFs calculated from simulation show similar behavior to the ACFs measured from experiment. Each of the 8 quadrants of the spherical surface (4 on the top hemisphere, 4 on the bottom hemisphere) displays heterogeneity in the amplitude of correlations because some quadrants contain large lattice fragments, and others contain mostly empty space (*Figure 7B*). The same trend is observed in a single VLP measured using super-resolution microscopy imaging (*Figure 7E*). Our simulations further show that as the lattice is stabilized, the ACF increases in amplitude and decays more slowly, which is qualitatively the same as is observed in imaging of Gag lattices in budded VLPs that have been stabilized with a fixative (*Saha and Saffarian, 2020*; *Figure 7C and F*). We cannot quantitatively compare the ACFs, as the experiments produced ACFs with much higher amplitudes of correlations, and even with the background correlations divided out (*Figure 7D*), the experimental signal contained additional sources of correlation likely due to measurement noise. However, we were able to use our simulations to illustrate how sources of measurement noise in stochastic localization imaging experiments can produce increased correlations beyond the background. We specifically find that short-term blinking of the fluorophore does not appreciably change the ACF (*Figure 7—figure supplement 1*). However, we do see increased amplitude of correlations in the ACF if we introduce a distribution of activation probabilities for the fluorophores, mimicking the fact that the populations initially activated may have a higher probability of activation than those appearing at later times (*Figure 7—figure supplement 2*). Further, if we assume that the lattice is not perfectly centered with respect to the activating laser pulses, then Gag monomers that are initially 'dark' can diffuse into view and then have a probability of being activated (*Figure 7—figure supplement 2*). This increases fluctuations in both the background signal and the signal from the separate quadrants. Thus, the simulations improve interpretation of the experiment, and more vividly bring to life the lattice dynamics and heterogeneity.

## Discussion

Our simulations of Gag lattices across a range of interaction strengths and rate constants all demonstrate how the incomplete lattice supports dynamic unbinding, diffusion, and rebinding of Gag along its fragmented edge. These dynamics and the resultant accessibility of Gag molecules along the edge of the lattice are important for activation of proteases via dimerization, and ultimately the maturation of the virion from this spherical lattice shell to the mature capsid. By measuring the FPT for dimer formation between a pair of Gag-Pol monomers, we show that dimerization can proceed in less than a few minutes, and for less stable lattices much faster, despite the embedding of these molecules within the lattice. By comparison with experimental measurements of lattice structure (via cryoET) and lattice dynamics (via biochemistry and time-resolved imaging), we conclude that the stability of the hexamer contacts should be in the range of $-10k_{\mathrm{B}}T < \Delta G_{\mathrm{hex}} < -8k_{\mathrm{B}}T$ for binding rates that are slower than $10^5$ M$^{-1}$s$^{-1}$. If the binding rates are faster, then the free energy could be further stabilized ($\Delta G_{\mathrm{hex}} < -10k_{\mathrm{B}}T$), as the dimerization events would still be fast enough to be consistent with the biochemical measurements. If the lattice is less stable than $-6k_{\mathrm{B}}T$, we found that the large-scale structure of the lattice is not maintained even within seconds, which is not consistent with structural measurements, and between $-6k_{\mathrm{B}}T$ and $-8k_{\mathrm{B}}T$, the dimerization is likely too fast relative to the biochemical measurements, although it is feasible given that our simulations predict an upper bound. These hexamer-hexamer contact strengths report the stabilities that would be expected given the

presence of co-factors, as without co-factors the lattice does not assemble at all (*Bush and Vogt, 2014*; *Campbell et al., 2001*), and co-factors are present in all experiments used for comparison.

Our simulations also demonstrate that during assembly, the fraction of Gag-Pol monomers, while only 5%, is still too high to prevent stochastic dimerization events between them. This means that preventing early activation, which can result in loss of proteases from the virion (*Bendjennat and Saffarian, 2016*) and significant reduction in virion formation (*Kräusslich, 1991*), requires active suppression of the interaction between adjacent Gag-Pol monomers. This suppression could occur in the form of highly unfavorable dimerization events between Gag-Pol monomers, which we found could significantly reduce the number of adjacent pairs, particularly if the assembly process allows for unbinding and 'correction' of such unfavorable contacts. Suppression could also occur by having adjacent pairs that are somehow enzymatically inhibited. The exact mechanism is not known. Ultimately, the suppression must be relieved to allow for protease activity in the budded virion, and our results show that two protease domains will be able to find one another even if seemingly locked within the lattice at distant locations.

Our model explicitly accounts for the crowding effects of localizing the lattice to a small, 2D surface, ensuring that excluded volume is maintained between all monomers. However, we do not explicitly include the gRNA that would be packaged within the immature virion and attached to the Gag lattice (through non-competing binding sites). It is known that binding to RNA (*Jouvenet et al., 2009*; *Rein et al., 2011*), membrane, and other co-factors is important in stabilizing the lattice for assembly (*Mallery, 2021*; *Kucharska et al., 2020*; *Dick et al., 2018*; *Datta et al., 2007*; *Webb et al., 2013*; *Duchon et al., 2021*; *Nikolaitchik et al., 2021*; *Lei et al., 2023*; *Sarni et al., 2020*). IP6 has been shown to accelerate and stabilize immature lattice assembly in vitro (*Kucharska et al., 2020*) and in vivo (*Mallery, 2021*). Our Gag-Gag interaction free energies thus presuppose that RNA and IP6 have bound already, as otherwise the lattice would not have assembled productively. Because we do not explicitly incorporate IP6 binding throughout the lattice, however, we are assuming it uniformly affects the lattice, whereas it could locally stabilize only where it is bound. IP6 is highly abundant in cells (~50 μM), and visible in cryo structures of the immature lattice (*Pak et al., 2022*), so it is likely that the majority of hexamers are interacting with IP6, but in future work it will be important to confirm this explicitly. Our Gag monomers and oligomers diffuse along the membrane surface, not through the interior of the budded virion where the RNA would be packaged, consistent with excluded volume in the virion center. When our lattice coverage changes from 33% to 66%, for example, we see only small changes in our MFPTs which primarily reflect the increase in total Gag-Pol monomers available. However, the attachment of the Gag lattice to a large RNA polymer of 9600 nucleotides (~3 μm) could change the mobility of the Gag monomers following their detachment from the lattice. Proteins can still unbind and diffuse when bound to a polymer-like RNA (*Pak et al., 2017*; *Elrad and Hagan, 2010*), but the effective rates could slow, and the distance that a monomer typically travels can be limited by the fluctuations of the attached RNA polymer. Hence, rebinding may be more restricted to shorter excursions from the start point. We note the Gag VLPs contain smaller RNA polymers and not the full gRNA, so the model is in that way more consistent with the dynamics of a VLP. Somewhat remarkably, the Gag monomers within the virion are able to reassemble around the gRNA to form the mature conical capsid (following cleavage) (*Sundquist and Kräusslich, 2012*), which indicates there is a clear capacity for diffusion driven remodeling. This mature lattice is also subsequently disassembled (*Márquez et al., 2018*), and the principles of our model here indicate how destabilization of hexamer contacts could help promote disassembly.

Our models here contain pairwise interactions, and cooperativity enters only in that the formation of a completed cycle (whether a hexamer or a higher-order cycle of multiple hexamers) is significantly more stable, because it requires two bond breaking events. However, coordination of the hexamer by IP6 can produce conformational changes (*Campbell et al., 2001*) or kinetic effects (*Pak et al., 2022*) that could change the stability of hexamer contacts between, say, a dimer vs a 5-mer. We did not include this additional cooperativity to keep the model as simple as possible; we expect that added cooperativity in hexamer formation would change the pre-factors in the quantitative relationship we predict between the hexamer free energies and the FPTs, as intermediates would be biased away from smaller fragments. However, because the lattice would inevitably still have the 'dangling' edges and partial hexamers observed experimentally (*Tan et al., 2021*), we would still see dissociation, diffusion, and rebinding events. Our model also does not incorporate any mechanical energy,

so while we capture local changes in stability due to defects in the lattice that reduce the protein contacts and thus free energy, we cannot measure directional forces or stresses within our lattice. Inhomogeneities in assembled lattices, like pentamers vs hexamers, result in varying mechanical stress (*Zandi and Reguera, 2005*), and defects or 'scars' in lattices on curved surfaces are known to represent mechanical weak points that are susceptible to cracking or fragmenting (*Negri et al., 2015*). This will be a particularly important extension for coupling the lattice with the mechanical bending of the membrane, which can be performed using continuum models (*Fu et al., 2021*). Lastly, other proteins are packaged into HIV-1 virions, including curvature inducers (*Inamdar et al., 2021*), and like RNA, additional protein interactions could shift the Gag unbinding kinetics. Ultimately, however, our models clearly show that despite the significant amount of protein-protein contacts and ordered structure within the membrane attached Gag lattice, there is nonetheless enough disorder along the incomplete edge to support multiple unbinding and rebinding events over the seconds to minutes timescale (*Video 1*, *Video 3*).

Although our work here is focused on the HIV-1 immature lattice, our approach could be insightfully applied to other retroviruses, particularly given the morphological differences between the closely related HIV-1 and HIV-2 immature lattices (*Martin et al., 2016*). The HIV-2 Gag polyprotein similarly forms the immature lattice at the plasma membrane, but imaging of the budded virion shows that the HIV-2 lattice is largely complete with an average membrane coverage ratio of 76%±8% (*Martin et al., 2016*; *Talledge et al., 2022*). Hence, although this lattice contains defects and gaps, it does not have the large vacancy present in the HIV-1 lattice studied here. Given the important role that this incomplete edge played in facilitating unbinding and rebinding events of Gag-Pol, we would expect that the protease dimerization events would be significantly slowed in the HIV-2 lattice. With higher surface coverage, the concentration of Gag-Pol is overall higher in the virion, which would help promote dimerization, but with less access to a long, incomplete edge, the number of un(re)binding events would be reduced. We found here that lattices that were initially assembled into 2–3 fragments rather than a single continent would have less of a large vacancy on the surface and exhibited slightly slower remodeling dynamics and increased FPTs for Gag-Pol dimerization. Ultimately, the HIV-2 lattice does still need to be cleaved and reassembled into the mature capsid, just like HIV-1 (*Martin et al., 2016*), so we would hypothesize that the binding kinetics between Gag contacts would have to be faster, to more readily promote the remodeling needed both for protease dimerization and the cleavage and disassembly of the immature lattice. Currently, there is significantly less detail on the assembly and maturation of HIV-2, and future research will be essential to gain a more comprehensive understanding of protease dimerization, activation, and maturation across various retroviruses.

Overall, the model and simulations here reveal a level of detailed Gag dynamics coupled to structural changes that are inaccessible to any single experiment but can nonetheless be compared to a range of experimental observables, as we have done here. Although diffusion does influence the collective dynamics of the lattice, for example, we find it does not significantly influence activation rates, as those are limited by binding and unbinding events rather than mobility. By defining a formula that allows us to extrapolate our model to other rates and free energies, we can predict how mutations that would change the strength or kinetics of the hexamer contacts would impact the timescales of the initial protease dimerization event. MFPTs can be predicted from theory in surprisingly complex geometries (*Bénichou et al., 2010*), but for the immature lattice, the problem is intractable without using simulation data due to the ability of Gag-Pol to rebind or 'stick' back onto the lattice through multiple contacts before successful dimerization encounters. More generally, modeling stages of viral assembly has been critical for establishing the regimes of energetic and kinetic parameters that distinguish successful assembly from malformed or kinetically trapped intermediates, such as in viral capsid assembly (*Endres and Zlotnick, 2002*; *Hagan, 2014*; *Grime et al., 2016*; *Jones et al., 2021*). Computational models of self-assembly can be used to assess how additional complexity encountered in vivo, such as macromolecular co-factors (*Pak et al., 2022*; *Pak et al., 2017*; *Mohajerani et al., 2022*), crowding (*Grime et al., 2016*; *Smith et al., 2014*), and changes to membrane-to-surface geometry (*Guo et al., 2022*), could help to promote or suppress assembly relative to in vitro conditions. Our reaction-diffusion model developed here provides an open-source and extensible resource (*Varga et al., 2020*) to study preceding and following steps in the Gag assembly pathway (as done in recent work [*Qian et al., 2023*]) with the addition of co-factors. A model of mature capsid assembly, for example, would involve Gag monomers that have a modified interface geometry and orientations

relative to one another, as quantified above. With rates and energies that match biochemical measurements, the model can act as a bridge between in vitro and in vivo studies of retroviral assembly and budding, and a tool to predict assembly conditions that disrupt progression of infectious virions.

## Methods

### Model components and structural details

Our model contains Gag and Gag-Pol monomers enclosed by a spherical membrane. The membrane contains binding sites for the Gag monomers. The Gag-Pol is structurally identical to the Gag but represents 5% of the total monomer population to track protease locations within the lattice. The model captures coarse structure of the Gag/Gag-Pol monomers as derived from a recent cryoET structure (*Schur et al., 2016*) of the immature lattice (*Figure 1A*). The key features of our rigid body models are the locations of the four binding sites/domains that mediate protein-protein interactions between a pair of Gag monomers and the Gag-membrane interaction. Each Gag/Gag-Pol contains a membrane binding site, a homo-dimerization site, and two distinct hexamer binding sites that support the front-to-back type of assembly needed to form a ring. When two molecules bind via these specific interaction sites, they adopt a pre-defined orientation relative to one another (*Varga et al., 2020*) that ensures the lattice will have the correct contacts, distances between proteins, and curvature (*Figure 1* and *Figure 1—source data 1*). The Gag monomers bind to the membrane from the inside of the sphere, as would be necessary for budding, and we model this as a single binding interaction that captures stabilization from $PI(4,5)P_2$ binding and myristolyation (*Ono et al., 2004*; *Saad et al., 2006*). Each reactive site excludes volume from only its reactive partners at a distance $\sigma$. The dimer site reacts with another dimer site at a binding radius of $\sigma$=2.21 nm. The MA site binds to the membrane at $\sigma$=1 nm. The hexamer site 1 binds to hexamer site 2 at $\sigma$=0.42 nm (*Figure 1B*). Once reactive sites have bound to one another, they are no longer reactive and no longer exclude volume. Therefore, to maintain excluded volume between monomers throughout the simulation, we introduce an additional dummy reaction between the monomer centers-of-mass (COM). The COM sites exclude volume with a binding radius of $\sigma$=2.5 nm between all monomer pairs. This is necessary to prevent monomers from unphysically diffusing 'through' one another when their reactive sites are fully bound.

### Reaction-diffusion simulations

Computer simulations are performed using the NERDSS software (*Varga et al., 2020*). The software propagates particle-based and structure-resolved reaction-diffusion using the free-propagator reweighting algorithm (*Johnson and Hummer, 2014*). The membrane is treated as a fixed continuum surface that contains a population of specific lipid binding sites, or $PI(4,5)P_2$. We model these binding sites using an implicit lipid algorithm that replaces explicit diffusing lipid binding sites with a density field that will change with time as proteins bind or unbind from the membrane. Hence, $PI(4,5)P_2$ are assumed well mixed on the surface. This method reproduces the kinetics and equilibria as the explicit lipid method but is significantly more efficient (*Figure 1—figure supplement 1*; *Fu et al., 2019*). We use a time-step $\Delta t$=0.2 $\mu$s. We validated the model kinetics as described in the next section. Software is open source here, https://github.com/mjohn218/NERDSS, and executable input files for the models are here, https://github.com/mjohn218/NERDSS/tree/master/sample_inputs/gagLatticeRemodeling.

We briefly describe here how the stochastic reaction-diffusion simulations work. Each protein or protein complex moves as a rigid body obeying rotational and translational diffusive dynamics using simple Brownian updates, for example $x(t + \Delta t) = x(t) + \sqrt{2D\Delta t}R$, where $D$ is the diffusion constant of the rigid body and $R$ is a normally distributed random number with mean 0 and standard deviation 1. Each protein binding site is a point particle that can react with a site on another molecule to define the reaction network, as illustrated by the contacts in *Figure 1*. Reactions can occur upon collisions, with the probability that the reaction occurs evaluated using the Green's function for a pair of diffusing sites, parameterized by an intrinsic reaction rate $k_a$, a binding radius $\sigma$, and the sum of the diffusion constants of both species (*Johnson and Hummer, 2014*). This reaction probability is corrected for rigid body rotational motion (*Johnson, 2018*). For proteins that are restricted to the 2D membrane, they perform 2D association reactions with 2D rate constants (*Yogurtcu and Johnson, 2015*), which are derived from the 3D rate constants by dividing out a length-scale $h$ that effectively captures the fluctuations of the proteins when on the membrane,

$$k_a^{2D} = k_a^{3D}/h \tag{2}$$

Proteins that do not react during a time-step undergo diffusion as a rigid complex, and excluded volume is maintained for all unbound reactive sites at their binding radius $\sigma$ by rejecting and resampling displacements that result in overlap. All binding events are reversible, with dissociation events parameterized by intrinsic rates $k_b$ that are sampled as Poisson processes. We have that for each reaction, $K_D = \frac{k_b}{k_a}$, and for the corresponding 2D reaction, we assume the unbinding rates are unchanged, and thus $K_D^{2D} = hK_D^{3D}$.

Binding interactions are dependent on collisions between sites at the binding radius $\sigma$ and are not orientation dependent. Orientations are thus enforced after an association event occurs by 'snapping' components into place. Association events are rejected if they generate steric overlap between components of two complexes. Steric overlap is determined using a distance threshold, where here if the distance between molecule COM is less than 2.3 nm, we reject due to overlap. They are rejected if they generate large displacements due to rotation and translation into the proper orientation, using a scaling of the expected diffusive displacement of 10. Defects ultimately emerge in the lattice because a hexagonal lattice cannot perfectly tile a spherical surface by the Euler polyhedron formula. These defects result in contacts that are not perfectly aligned (*Figure 2*); if the contacts are within a short cutoff distance of $1.5\sigma$, they can still form a bond to stabilize the local order, otherwise they are left unbound, weakening the local order.

## Transport

We estimate translational ($D$) and rotational ($D_R$) diffusion coefficients from the Einstein-Stokes equations, assuming a higher viscosity for an in vivo process. We define for a Gag in solution: $D_{Gag}(D_{Gag-Pol})$=10 $\mu m^2$/s, $D_{R,Gag}(D_{R,Gag-Pol})$=0.01 rad$^2$/$\mu s$. Membrane $D_{lipid} = 0.2$ $\mu m^2$/s. Diffusion slows as complexes grow, consistent with a growing hydrodynamic radius and quantified by the Einstein-Stokes equation (*Varga et al., 2020*). Hence, a single protein on the membrane diffuses at $1.96\times10^{-1}$ $\mu m^2$/s, and a membrane bound complex containing 1000 proteins diffuses at $1.96\times10^{-4}$ $\mu m^2$/s.

## Energetic and kinetic parameters

We studied lattice dynamics at several strengths defining the free energy $\Delta G_{hex}$ of the hexamerization interaction, at $-5.62k_BT$, $-7.62k_BT$, $-9.62k_BT$, $-11.62k_BT$, where $k_B$ is Boltzmann's constant and $T$ is the temperature. The Gag and Gag-Pol have identical energetic and kinetic parameters to one another during all remodeling simulations. We specified the dimerization free energy $\Delta G_{dimer}$ at $-11.62k_BT$ and $-13.62k_BT$, which straddles the stability of the measured solution $K_D$ of 5.5 μM or $-12.1k_BT$ (*Datta et al., 2007*). Additional stabilization of the dimer interaction can accompany conformational changes (*Datta et al., 2011*), which could drive stronger Gag-Gag binding within the lattice (*Datta et al., 2007*). Given the $\Delta G$ values ($\Delta G = G_{bound} - G_{unb}$) and using, $K_D = c_0 \exp\left(\frac{\Delta G}{k_BT}\right)$, where $c_0$ is the standard state concentration (1 M), we further selected a set of on- and off-rates at each free energy, where we used both hexamer and dimer intrinsic binding rates $k_a^{2D}$ of $2.5\times10^{-3}$ nm$^2$/μs, $2.5\times10^{-2}$ nm$^2$/μs, and $2.7\times10^{-1}$ nm$^2$/μs. Off-rates are constrained by $\Delta G$ via $K_D = k_b/k_a^{3D}$, $k_a^{2D} = k_a^{3D}/h$. For the Gag-membrane interaction, $h$=2 nm, for the Gag-Gag interactions, $h$=10 nm, comparable to the size of the Gag monomer.

Our model allows for a strain energy in the formation of closed polygons-like hexagons. We set that energy $\Delta G_{strain}$ here to $+2.3k_BT$ for all models, meaning that the stability of any closed hexagon within the lattice is slightly lower compared to 6 ideal bonds (i.e. for $\Delta G_{hex} = -11.62k_BT$ it is 5.8 bonds) but still much more stable than a linear arrangement of six Gag monomers which has only 5 bonds. This is an entropic penalty to forming closed cycles which require the final subunit to fit into the 5-mer structure and form 2 bonds simultaneously. This could mechanistically result from compressed or stretched arrangement of subunits in the hexameric cycles on the curved membrane, compared to a more favorable spacing in solution where no forces from the membrane exist. This penalty only affects the lifetimes of the hexamer cycles. When a hexamer closes to form a cycle, it forms 2 bonds and thus has a stability of $K_{D,cycle} = c_0 \exp\left(\frac{2\Delta G_{hex}}{k_BT}\right) \exp\left(\frac{\Delta G_{strain}}{k_BT}\right) = K_D \exp\left(\frac{\Delta G_{hex}+\Delta G_{strain}}{k_BT}\right)$, where $\Delta G_{hex} < 0$ and a penalizing $\Delta G_{strain}$ is >0. We perform association reactions with the same forward rate, which means that $k_{b,cycle} = k_a K_{D,cycle} = k_b \exp\left(\frac{\Delta G_{hex}+\Delta G_{strain}}{k_BT}\right)$. We do not apply this strain penalty to dimer bonds that can also end up in higher-order cycles, thus assuming that they can accommodate

spacing or small structural rearrangements without any free energy cost. We ran a set of comparison simulations where $\Delta G_{\text{strain}} = 0$, to illustrate how it can impact the structures of the weaker lattices. Quantitatively, setting $\Delta G_{\text{strain}}$ to 0 gives the hexamer cycles a lifetime that is 10-fold longer. For $\Delta G_{\text{hex}} = -11.6 k_{\text{B}}T$, the hexamer lifetime thus increases from 1380s to 13,800 s when $k_{\text{a}}^{\text{3D}}=1.5\times10^5$ M$^{-1}$s$^{-1}$, but these are both dramatically slower than a single hexamer bond which has a lifetime of 0.74 s. For the weakest lattice, however, the hexamer cycle is only 4× more long-lived than a single bond, so the strain penalty is more impactful. With additional dimer interactions stabilizing subunits in the lattice, however, hexamer lifetime increases further.

We validated the kinetics and equilibrium of our model as it assembled on the membrane when we set the hexamer rates to 0, so it formed purely dimers (*Figure 1—figure supplement 1*), and when we set the dimer rates to 0, so it formed purely hexamers (*Figure 1—figure supplement 2*). The observed kinetics and equilibria were compared to solutions solved using the corresponding system of non-spatial rate equations, showing very good agreement with apparent intrinsic rates that systematically accounted for excluded volume and the criteria used for accepting association events (*Figure 1—figure supplement 1*). Thus, all of the rates and free energies reported in the paper agree with the kinetics and equilibria observed and expected for the sets of binding interactions that make up the full lattice system.

## Simulations for constructing the initial lattices on the membrane

The HIV lattice is composed of Gag and Gag-Pol bound to the inner leaflet of the lipid membrane. To study the remodeling dynamics, we must construct the initial configurations where the lattice is assembled such that it has a specific coverage of the surface (67% or 33%), and is linked to the membrane via lipid binding. We define the membrane sphere of radius 67 nm to represent the membrane surface (*Sundquist and Kräusslich, 2012*). PI(4,5)P$_2$ is populated on the membrane surface at a concentration 0.07 nm$^{-2}$, or 4000 copies, which exceeds the number of Gag monomers, meaning there is always a pool of free PI(4,5)P$_2$ available for (re)binding.

Assembling the Gag monomers into a single spherical lattice is non-trivial due to the size of the lattice. Because the lattice is so large, requiring *N*~2400 monomers at 67% coverage, it is very difficult for a single nucleated lattice to complete growth (which scales approximately with *N*) before another lattice nucleates. These multiple intermediate fragments do not readily combine. In a recent study we quantified how titrating in monomers instead of trying to assemble from the bulk can dramatically improve assembly yield (*Qian et al., 2023*). Therefore, here we titrate in the Gag and Gag-Pol monomers at a rate of 6×10$^{-5}$ M/s and 3×10$^{-6}$ M/s respectively, which can ensure a ratio of Gag:Gag-Pol of ~20:1, consistent with experiment (*Sundquist and Kräusslich, 2012*; *Garcia-Miranda et al., 2016*). Gag(-Gag-Pol) molecules can bind in solution (3D), to the membrane (3D to 2D), and when on the membrane (2D). In one set of assembly simulations we set binding rates between Gag-Pol and Gag-Pol pairs to 0 to try and suppress the 'activation' events that could therefore occur during assembly (*Figure 2*). While the titration of the monomers reduced multiple nucleation events for the membrane system, we found that the easiest and most efficient way to form a single lattice was by assembling the structure fully in solution, in a volume of (250 nm)$^3$. We then put the assembled single lattice into a spherical system by linking this structure to the membrane using one PI(4,5)P$_2$ attachment per monomer. The Gag rates of dimerization and hexamerization were both set to 6×10$^6$ M$^{-1}$s$^{-1}$. For the lattices studied below, we made binding events irreversible, as it improved the growth of single lattices. For comparison, we also ran a few assembly simulations where the binding was reversible, using $\Delta G_{\text{hex}} = -11 k_{\text{B}}T$ and $\Delta G_{\text{dimer}} = -13 k_{\text{B}}T$, $k_{\text{off}} = 100$ s$^{-1}$ and 13.6 s$^{-1}$, respectively. These reversible binding simulations also used titration, and although they often nucleated two structures, we could keep adding monomers until at least one lattice reached our target size. Because the hexamer and dimer rates are identical during the assembly process, we do not see selection for only complete dimers along the lattice periphery, as is observed in the cryoET maps (*Tan et al., 2021*). To recover this feature, we would instead need to assemble the lattice under more native-like conditions where the dimer is more rapidly and stably formed compared to the hexamer contact. We generated 16 initial configurations for each coverage area (67% and 33%). Some initial configurations are shown in *Figure 2*.

## Simulations for lattice remodeling dynamics

For each initial configuration we have generated, we perform six independent trajectories. See *Video 1* for one trajectory. We perform these 96 simulations for each set of model parameters to

**Table 1.** Simulation parameters for remodeling dynamics.

| | | | |
|---|---|---|---|
| Gag copy number | ~2500 | | |
| Gag-Pol copy number | ~125 | | |
| Lipid copy number | 4000 | | |
| Radius of sphere | 67 nm | | |
| Time-step | 0.2 µs | | |
| $k_a^{2D}$ (nm²/µs), $k_a^{3D}$ (M⁻¹s⁻¹), $k_b$ Gag-Mem | 1, 1.2×10⁶, 0.61 s⁻¹ | | |
| $k_a^{2D}$ (nm²/µs) Gag-Gag dimer | 2.5×10⁻³ | 2.5×10⁻² | 2.58×10⁻¹ |
| $k_a^{2D}$ (nm²/µs) Gag-Gag hexamer | 2.5×10⁻³ | 2.5×10⁻² | 2.737×10⁻¹ |
| $k_a^{3D}$ (M⁻¹s⁻¹) Gag-Gag hexamer | 1.5×10⁴ | 1.5×10⁵ | 1.6×10⁶ |
| $k_b$ Gag-Gag dimer (s⁻¹) (−11.62$k_B T$) | 1.35×10⁻¹ s⁻¹ | 1.35×10⁰ | 1.4×10¹ |
| $k_b$ Gag-Gag dimer (s⁻¹) (−13.62$k_B T$) | 1.8×10⁻² s⁻¹ | 1.8×10⁻¹ | 1.89×10⁰ |
| $k_b$ Gag-Gag hexamer (s⁻¹) (−5.62$k_B T$) | 5.45×10¹ | 5.5×10² | 6.01×10³ |
| $k_b$ Gag-Gag hexamer (s⁻¹) (−7.62$k_B T$) | 7.37×10⁰ | 7.44×10¹ | 8.13×10² |
| $k_b$ Gag-Gag hexamer (s⁻¹) (−9.62$k_B T$) | 1.0×10⁰ | 1.0×10¹ | 1.1×10² |
| $k_b$ Gag-Gag hexamer (s⁻¹) (−11.62$k_B T$) | 1.35×10⁻¹ | 1.36×10⁰ | 1.49×10¹ |
| $\Delta G_{strain}$ | 2.3$k_B T$ | | |

generate statistics both within and across initial configurations. For some simulations, fragments of the lattice become sterically overlapped with one another, due to the high density and the time-step size. While this could be eliminated by lowering the time-step, we instead keep the more efficient time-step, and discard these simulation traces which produce overlap. We finally analyze 60 remodeling traces for each parameter set. All the simulation parameters are listed in *Table 1*. The number of monomers is fixed for each simulation by the initial configuration, so that only binding, unbinding, and diffusion can occur throughout the simulation. During lattice construction, in one set of simulation we set all Gag-Pol to Gag-Pol binding interactions to zero (*Figure 2B*). Now for the remodeling dynamics, we allow all interactions involving Gag-Pol, and at rates that are identical to those involving Gag, meaning there is no difference between the types except for in their label. The Gag/Gag-Pol molecules are allowed to diffuse on the membrane, where they can unbind from a molecule and rebind to another with the specified binding rates. Each monomer can also unbind and rebind to the membrane lipids. However, dissociation to solution is extremely rare, as it requires that all Gag monomers in an assembled complex unbind from their lipid before any of the sites rebind. In *Figure 1—figure supplement 4*, we confirm that even for the most unstable lattice that produces small mobile fragments, the dynamics are very similar when the lipid binding is modeled using implicit or explicit binding sites.

## Calculation of First-passage times (FPT)

Our primary observable is how long it will take for the first dimerization event between a pair Gag-Pols Our simulations are stochastic and thus this is an FPT measurement (*Iyer-Biswas and Zilman, 2016*). The 'clock' is started from the initialized lattices as shown in *Figure 2*. We note that two Gag-Pols have a chance to be adjacent at the initial configuration (*Figure 2*), but we ignore these events since dimerization of Gag-Pol before viral release has been experimentally shown to result in loss of Pol components from the virions (*Bendjennat and Saffarian, 2016*).

## Calculation of FPTs

Our primary observable is how long it will take for the first dimerization event between a pair Gag-Pols. Our simulations are stochastic and thus this is a first-passage time measurement (*Iyer-Biswas and Zilman, 2016*). The 'clock' is started from the initialized lattices shown in *Figure 2*. We note that two Gag-Pols have a chance to be adjacent at the initial configuration (*Figure 2*), but we ignore these

events since dimerization of Gag-Pol before viral release has been experimentally shown to result in loss of Pol components from the virions (*Bendjennat and Saffarian, 2016*).

## Fitting of the MFPTs

Given our distribution of FPTs calculated across our 60 trajectories per model, we can calculate the MFPT. To derive a phenomenological expression that captures our measured MFPT, we used a single global formula for all our model results:

$$\tau = a_1 \left( k_a^{2D}/\mathrm{SA} \right)^{-1} \exp \left( a_2 \Delta G_{\mathrm{hex}}/k_B T \right) \qquad (3)$$

where there are two fit parameters, $a_1$ and $a_2$, and the surface area $\mathrm{SA} = 4\pi R^2$ of the sphere is the same for all models. This expression is inspired by characteristic timescales for bimolecular association, which are inversely dependent on the reaction rate (here $k_a^{2D}$) (*Mishra and Johnson, 2021*). The $SA$ is present to ensure the correct units for $\tau$, and we empirically observe the relationship between the hexamer free energy and the measured MFPT. We optimized the parameters $a_1$ and $a_2$ using nonlinear fitting in MATLAB to the $\ln(\tau)$ functional form of *Equation 3*.

## Calculation of binding timescales from biochemical experiments

Recent measurements on Gag VLPs quantified dimerization times between sub-populations of tagged Gag molecules within the immature lattice (*Saha et al., 2021*). Dimerization events were identifiable because one sub-population of Gag monomers carried a HALO-tag (a protein that fuses to a target of interest, here Gag), and another carried a SNAP-tag. The addition of a linker HAXS8 produced a covalent linkage which we will call HALO-link-SNAP. The concentration of these HALO-link-SNAP structures was then quantified vs time. We therefore reproduced this experiment via analysis of our simulation trajectories. We defined a population of our Gag monomers randomly selected to have a 'SNAP-tag' and a population randomly selected to have a 'HALO-tag'. For each trajectory, the tagged populations of each were either 5%, 10%, 20%, or 40% of the total monomers, to match the experimental measurements (*Saha et al., 2021*). We then monitored the number of dimerization events that occurred as a function of time. A dimerization event required that a monomer with a SNAP-tag and a monomer with a HALO-tag encountered one another at a distance less than 3 nm (*Erhart et al., 2013*), where one and only one of these partners must have the covalent linker attached. Three nm cutoff distance is comparable to the molecular length-scales of the two protein tags with the linker between them (*Erhart et al., 2013*). We randomly selected half of the population of SNAP-Gags to have a linker attached, and half of the population of HALO-Gags to have a linker attached, and thus some encounters between a SNAP- and HALO-Gag were not productive if 0 or 2 linkers were present. Ultimately, however, all dimers could be formed given the symmetric populations containing linkers. These binding events were irreversible, consistent with a covalent bond formed.

This model assumes that the arrival of the linker to the inside of the virion is relatively rapid. The permeability coefficient of the linker when exposed to the membrane enclosed Gag lattice is approximately 0.0004 nm/μs (*Erhart et al., 2013*), and assuming a membrane thickness of ~5 nm, the diffusion across the membrane occurs at ~0.002 nm²/μs. To test the role of linker permeability, we solved the diffusion equation for a 1 μM concentration of linker molecules diffusing into a sphere of radius $R$=67 nm, which mimics the experiments. Within 100 ms, the concentration of the linker at 60 nm (close to the Gag-tagged end) has already reached 0.8 μM. Hence, although there is some delay following addition of the linker, it is much less than the time (20 s to 3 min) over which most of the dimerization occurs. The model also assumes that the linker does not saturate all HALO and SNAP molecules independently, which would prevent any dimers forming. The rates of binding of SNAP and HALO to the linker HAXS8 are $3\times10^4$ and $3\times10^6$, respectively (*Erhart et al., 2013*). We solved a system of ordinary differential equations for binding of HALO and SNAP to a linker given these rates. The HALO and SNAP concentrations were controlled by the size of the virions with 250 of each present (10%), and the linker concentration was 1 μM, which was found experimentally to ensure high dimerization success (*Saha and Saffarian, 2020*). Although the linker binds more rapidly to HALO, there is still plenty of time for the HALO-linked molecules to bind to a free SNAP before all the sites are occupied by linkers, as the copy numbers of linkers in the volume are low. In particular, if the HALO

and SNAP tags are adjacent in the lattice, the SNAP is much more likely to bind the adjacent HALO-linker than a free linker.

Since our simulations are ~20 s, we did a linear fit of the last second of the dimer forming kinetics to extrapolate the dimer copies at 3 min, which can be used for comparison with the experiment. This extrapolation therefore assumes that dimer formation does not slow down, which it almost certainly does. All of our assumptions contribute to the maximal possible dimerization efficiency, and thus our observables provide an upper bound on the expected number of HALO-link-SNAP dimers.

## Calculation of number ACF

Recent experiments also measured an ACF of immature lattice dynamics using time-resolved super-resolution microscopy. These experiments on Gag VLPs used interferometric photoactivated localization microscopy (iPALM) to track collective motion of the Gag lattice by stochastically localizing individual monomers to precise locations in the lattice over several minutes (*Saha and Saffarian, 2020*). To mimic the experimentally extracted observable, we counted the number of Gag monomers found on each fixed 1/8 of the sphere surface. The copy numbers within each of the 8 quadrants vary due to diffusion of the lattice, while the total copy numbers on the surface is fixed. The copy numbers at a time point $t$ are denoted by $G(t)$, and thus the ACF for each quadrant is given by:

$$ACF(\tau) = \frac{\langle G(t+\tau) G(t) \rangle}{\langle G(t+\tau) \rangle \langle G(t) \rangle} \tag{4}$$

These ACF measurements are comparable to fluorescence correlation spectroscopy measurements (*Wohland et al., 2001*). For comparison, the ACF for $G(t)$ across the full sphere surface is 1 at all times, because there is no change in total copy numbers. At long times, when the counts are uncorrelated, this function will go to 1, because $\langle G(t+\tau) G(t) \rangle \to \langle G(t+\tau) \rangle \langle G(t) \rangle$. Furthermore, if the copies are all well mixed across the surface, then we expect minimal deviations from 1 across all times given these relatively large viewing regions (1/8 of the surface), because there is no source of correlation between copies if they are well mixed. As $\tau \to 0$, the deviation of this ACF from 1 reports on the variance of the fluctuations in the numbers per patch relative to the mean, or the coefficient of variation (CV) squared: $ACF(\tau=0) = \frac{\langle G(t)^2 \rangle - \langle G(t) \rangle^2}{\langle G(t) \rangle^2} + 1 = \frac{\sigma^2}{\mu^2} + 1$. We calculated the number correlation for each quadrant of one simulation trace. We then took the average of all 60 traces for each parameter set. We also used an ensemble averaging approach, where we calculated the copy number correlations and means across multiple quadrants and trajectories before averaging to get the numerator and denominator of *Equation 4*. This method provides more statistics on longer time delays, and is based on assuming that all trajectories are sampling from the same equilibrium distribution. The trends are the same as those we report but shifted up to slightly higher amplitudes before decaying to 1. We note that our ACFs are not truly reporting on equilibrium fluctuations, as we show below that there is clearly some time-dependent changes to the lattice structure that is not reversible due to fragmenting along the edge compared to the initial structures. However, we compared the ACFs calculated for the first half of our simulations vs the last half, for example, and all of the same trends are preserved.

In addition to directly calculating this number autocorrelation, we also sampled it using a stochastic localization approach that directly mimics the experimental measurement (*Saha and Saffarian, 2020*). For this approach, for each time point we 'activate' a single Gag monomer across the full lattice with a probability $p_{act}$. So for each frame, either 1 or 0 Gag monomers is visible. We identify the quadrant for that monomer, and thus each quadrant produces a sequence of 1s and 0s. After a monomer is activated, it is then bleached, and cannot be localized again. The sequence of localizations is then used to calculate the same ACF, where we use a binning method (*Wohland et al., 2001*), as in the experimental analysis, to improve statistics on the signal at larger time delays. The agreement between the stochastic measurement of the ACF and the direct measurement of the copy numbers ACF are excellent (see Fig 7). We use this stochastic localization method so that we can introduce additional sources of correlation to our measurement of the simulated lattice dynamics, since these measurement artifacts can appear in the real experimental system.

## Analysis of ACF from experimental data on VLPs

The time-resolved microscopy (iPALM) experiments to characterize lattice dynamics in VLPs were previously described and published (*Saha and Saffarian, 2020*). We describe the analysis of these stochastic localization experiments here because we focus on analyzing a shorter part of the measurement. We analyzed only the first 500 s of the measurement (5000 frames), because after that time the laser intensity was changed. Based on data collected on 25 VLPs, we analyzed only the VLPs where they reported a large enough fraction of localization events to indicate a reliable measurement, so we included only VLPs where >75% of the quadrants had more than 250 localizations, leaving 11 VLPs. We used the same algorithm (*Wohland et al., 2001*) as applied to the simulation data to quantify the ACF from the time-dependent sequences of localization events (typically a series of 1s and 0s, with occasionally 2 events per frame). For each of the 8 quadrants, we effectively measure the copies of monomer per quadrant: $n_1(t)$, $n_2(t)$,... $n_8(t)$. The total copies are then $N(t)= n_1(t)+n_2(t)+... +n_8(t)$. The ACF for any single quadrant is given by $ACF\left(\tau\right) = \frac{\langle n_i\left(t+\tau\right)n_i\left(t\right)\rangle}{\langle n_i\left(t+\tau\right)\rangle\langle n_i\left(t\right)\rangle}$. For the total surface, $ACF\left(\tau\right) = \frac{\langle N\left(t+\tau\right)N\left(t\right)\rangle}{\langle N\left(t+\tau\right)\rangle\langle N\left(t\right)\rangle}$, which we denote as the background signal, as the full surface was visualized once per experiment, with localization events then assigned to quadrants. This background signal reports on fluctuations in the total copy numbers of Gag on the surface, which we would expect to be 1 given a perfect measurement, but which was always higher than this due to measurement noise. To remove this effect of total copy number variations, and instead focus on the local fluctuations in concentrations per quadrant, we would like to report a corrected $ACF_{\text{corr}}\left(\tau\right) = \frac{\langle n_i\left(t+\tau\right)/N\left(t+\tau\right)n_i\left(t\right)/N\left(t\right)\rangle}{\langle n_i\left(t+\tau\right)/N\left(t+\tau\right)\rangle\langle n_i\left(t\right)/N\left(t\right)\rangle}$. However, we do not have access to the relative concentrations $n_i\left(t\right)/N\left(t\right)$ at each time-step from experiment. A reasonable approximation is to assume that we can separate the average behavior of $n_i\left(t\right)$ and $N\left(t\right)$, such that $ACF_{\text{corr}}\left(\tau\right) \approx \frac{\langle n_i\left(t+\tau\right)n_i\left(t\right)\rangle/\langle N\left(t+\tau\right)N\left(t\right)\rangle}{\langle n_i\left(t+\tau\right)\rangle\langle n_i\left(t\right)\rangle/\langle N\left(t+\tau\right)\rangle\langle N\left(t\right)\rangle}$, which is equivalent to dividing out the background ACF from the signal of each quadrant. This background-corrected ACF reproduces the exact ACF when the total copy numbers are constant. We then averaged these signals across the VLPs. We performed the same analysis on the 25 VLPs that had been modified by a fixative, first filtering out the VLPs that had too few localization measurements (leaving 13 VLPs), and then otherwise proceeding with the analysis in an identical fashion.

## Acknowledgements

MEJ gratefully acknowledges funding from an NSF CAREER Award 1753174. The funders had no role in study design, data collection and analysis, decision to publish, or preparation of the manuscript. We acknowledge use of the ARCH supercomputer Rockfish at Johns Hopkins, with support from NSF MRI 1920103 and the XSEDE supercomputer Stampede2 through XRAC MCB150059. Contributions from IS and SS were supported by NIH R01 AI150474. We thank Prof. John Briggs for sharing the datasets defining the Gag hexamers within the immature Gag lattice from cryoET.

## Additional information

### Funding

| Funder | Grant reference number | Author |
| --- | --- | --- |
| National Science Foundation | 1753174 | Margaret E Johnson |
| National Institutes of Health | R01 AI150474 | Saveez Saffarian |

The funders had no role in study design, data collection and interpretation, or the decision to submit the work for publication.

### Author contributions

Sikao Guo, Resources, Data curation, Software, Formal analysis, Validation, Investigation, Visualization, Methodology, Writing - original draft, Writing - review and editing; Ipsita Saha, Data curation,

Methodology; Saveez Saffarian, Conceptualization, Data curation, Supervision, Methodology, Writing - review and editing; Margaret E Johnson, Conceptualization, Formal analysis, Supervision, Funding acquisition, Investigation, Methodology, Writing - original draft, Project administration, Writing - review and editing

**Author ORCIDs**
Sikao Guo ⬚ http://orcid.org/0000-0002-7680-8060
Margaret E Johnson ⬚ http://orcid.org/0000-0001-9881-291X

**Decision letter and Author response**
Decision letter https://doi.org/10.7554/eLife.84881.sa1
Author response https://doi.org/10.7554/eLife.84881.sa2

## Additional files

**Supplementary files**
• MDAR checklist

**Data availability**
All software used for simulations is available open source at https://github.com/mjohn218/NERDSS, (copy archived at *Guo et al., 2023*), and executable model files and instructions for running them are freely available here.

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
