## [Editor Report]

This fundamental work substantially advances our understanding of the maturation of retroviruses, a key step in understanding the formation of infectious viruses. The evidence supporting the conclusions is compelling, with rigorous computational simulations. The work will be of broad interest to the community of virologists worldwide.

---

## [Decision Letter]

**Decision letter after peer review:**

Thank you for submitting your article "Defects in the HIV immature lattice support essential lattice remodeling within budded virions" for consideration by *eLife*. Your article has been reviewed by 3 peer reviewers, one of whom is a member of our Board of Reviewing Editors, and the evaluation has been overseen by Miles Davenport as the Senior Editor. The reviewers have opted to remain anonymous.

Essential revisions:

1) Including the PIP2 and IP6 role on their implicit lipid membrane model.

2) Quantifying the similarity of the packaging defects of the immature lattice observed by the authors in their model and those seen experimentally from cryoET.

3) To confirm that the authors' coarse model is representative of immature CA and not mature CA.

4) The argument is made that HIV-1 'utilizes' defects in its lattice to accommodate the protease, providing access to cleavage sites. The authors state that complete, 100% coverage would occlude the cleavage site and prevent proteolysis. Add some discussion about how HIV-2 fits their suggested paradigm.

5) Results F, the MFPT formula is an empirical fit. The authors should qualify this by an appropriate measure of the quality of fit, such as an R^2^ measure adjusted for small sample sizes, for example.

6) Re-wording the current title.

*Reviewer #1 (Recommendations for the authors):*

The authors found that even in the incompleteness it is extremely unlikely that a lattice will be assembled without a pair Gag-Pols already adjacent (Figure 2B), an observation that clearly suggests that there must exist a mechanism of inhibition until the virion budding since, as stated by the authors, any activation preceding budding is known to reduce infectivity. So, wouldn't it be more interesting to focus on the possible role of the incompleteness of the lattice as a possible mechanism of Pol inhibition? Would it be possible to determine the number of Gag-Pol molecules that would stochastically be located adjacent but with lower-energy interactions which would favor their detachment and separation? How would this be affected by different levels of incompleteness? Would a more-dense assembly lead to Gag-Pol molecules being "trapped" without the possibility of avoiding activation? How would this different point of view be put in context with observations made by Tan A. et al. 2021?

The title seems a bit disconnected from the main body of the manuscript.

*Reviewer #2 (Recommendations for the authors):*

General: Can the authors elaborate on their implicit lipid membrane model and how PIP2 interacts with their protein model? The HIV liposome has an unusual composition, enriched in Cholesterol and Sphingomyelin, and its composition is asymmetric across leaflets. Further, lipid microdomains ostensibly form on the surface and even direct the localization of gag on the membrane surface. MA binds PIP2 which is a part of the maturation cascade, which the authors include as a "stabilization". Does the implicit model treat any of these points directly, and if so how? Where are PIP2 molecules located on the membrane surface? A supplemental figure may help clarify the latter point.

General: The claim is made that the packaging defects of the immature lattice are similar to those seen experimentally from cryoET. Is there a way to quantify this? On visual inspection alone, they don't seem similar to me. Natural lattice defects seem to be uniformly distributed along the capsid surface, whereas those stemming from this model appear to be a consequence of surface area (i.e., one large continuous region and a corresponding bare region). Some quantification or comparison against published data is necessary.

General: Immature CA has two dimerization sites, one in the N-terminal domain (helix 1) and another in the C-terminal domain (helix 9). Which dimerization site is treated with the virtual particle, purple, shown in Figure 1 panel B.

General – related to the above point: Immature and mature CA is related by differing relative orientations of the N- and C-terminal domains, and this further modulates intermolecular interactions between CA monomers, i.e., different dimerization, trimerization, etc. How do the authors confirm that their coarse model is representative of immature CA and not mature CA? Are there enough details in the model to delineate this feature?

General: Can the authors explain the source of curvature in their model? Does it come from the virtual dimerization site (purple bead in Figure 1 panel B)? Or is it directed by the implicit membrane?

General: What are the exact differences between Gag and Gag-pol in the simulations? It is clear from the text that there are parametric differences, but structurally this is not clear.

General: The cryoET model referenced in the paper does not include the MA domain. How did the authors position their MA bead, and by what criteria was this accomplished, aside from the condition of being normal to the membrane surface? For instance, how was the distance of the bead from the CoM chosen?

Lines 489-491: The authors suppress multiple nucleation events in the present work, and do not provide any rationale for this choice. It is the opinion of this reviewer that the authors should extend their calculations to include multiple nucleation events. Is there biological evidence that only a single nucleation event initiates spherical particle formation? This point is also alluded to on line 307 as a challenging aspect of this modeling endeavor. If it's real, why not allow it in your models?

Lines 530-539: The argument is made that HIV-1 'utilizes' defects in its lattice to accommodate the protease, providing access to cleavage sites. The authors state that complete, 100% coverage would occlude the cleavage site and prevent proteolysis. This begs the question about HIV-2, or other retroviruses, which is known to have a significantly more complete immature lattice. Can the authors add some discussion about how HIV-2 fits their suggested paradigm? This would be an interesting inclusion for readers.

Figure 4 panel C – According to the caption, the fits in this panel were taken from data points in which 75% of the trajectories produced dimerization events, which yields a linear fit of the data. Is it valid to take only those data points where 75% of the trajectories produced dimerization? Since if the additional data points are included, the trend changes completely and is non-linear. Some additional discussion about this choice would be helpful – in the caption or otherwise.

General: there is a lot of jargon, which makes the paper difficult to decipher for a general audience. Can the authors expand and/or elaborate where necessary, e.g., iPALM.

*Reviewer #3 (Recommendations for the authors):*

My main comments are summarized in the "public review". Privately, and despite the evident seriousness of this study, I found the results a little underwhelming. That is, there does not seem to be anything particularly surprising, or conceptually new. This, therefore, raises questions of significance, and how the collective impact of the results will serve to drive the HIV field forward, either by comprehensively solving an outstanding problem or by opening up a new avenue of research. Of course, there may be some context here that I have missed, but at present, I am struggling to see it.

---

## [Author Response]

Essential revisions:1) Including the PIP2 and IP6 role on their implicit lipid membrane model.

We have now run additional simulations with explicit lipids that confirm the accuracy of the implicit lipid model, and clarified in the text the properties of the implicit lipid model. In our simulations, the membrane is a continuum surface. The PIP2 lipids serve as binding sites for each Gag monomer. Both when they are explicitly modeled as diffusing sites, and implicitly modeled as a uniform density of sites, the proteins can bind and unbind to the sites, thus impacting the number of free and occupied sites in time. We use the implicit lipid model because it is significantly more computationally efficient than propagating individual lipid sites, although it sacrifices the full spatial resolution of lipid sites. To verify that the implicit lipid model does not impact our results, we compared the assembly kinetics for the most unstable lattice, ∆G_hex_=-5.62k_B_T. ka_2D_ (nm^2^/µs) = 2.5x10^-2^. This lattice has fast dynamics and the smallest mobile fragments, thus it is more sensitive to the kinetics of un/rebinding the membrane. We observe a very close match between the explicit and implicit lipid simulations as shown in the kinetics in a newly added Figure 1—figure supplement 3, indicating that the implicit lipid model can accurately represent lipids while offering significantly higher computational efficiency. (See revisions on pages 8, 35, and 45 of the manuscript). The implicit lipid model does show a slight shift in the average size of lattices, indicating that there is a small reduction in rebinding with explicit lipids, which will be due to their non-uniform distribution in space which is not accounted for in the implicit lipid model.

These deviations will become negligible for more stable lattices that have larger fragments and thus more simultaneous links to the membrane.

Regarding the role of IP6, we have clarified in the manuscript that the interaction rates and free energies we assign to the Gag-Gag interactions presuppose its presence, as IP6 is known to promote immature lattice assembly in vivo. We acknowledge that our current model thus assumes IP6 uniformly affects the lattice, rather than potentially only locally stabilizing the lattice where it is bound. IP6 is highly abundant in cells (~50uM), so it is possible that the majority of hexamers do indeed bind IP6; IP6 is visible in cryoEM structures. However, it would be important in future work to explicitly include the binding to IP6 and thus establish its local influence both on lattice assembly and stability (See revisions on page 26 of the manuscript). We are unaware of any influence IP6 directly has on membrane binding, in case we misinterpreted the comment.

2) Quantifying the similarity of the packaging defects of the immature lattice observed by the authors in their model and those seen experimentally from cryoET.

We have performed additional analysis of our simulated Gag lattices and the lattices constructed from cryoET, with experimental datasets provided by the Briggs group. In Figure 2C we now include images of our Gag lattices that color the complete hexamers vs the incomplete hexamers. We quantify that these lattice defects (incomplete hexamers) constitute 35-40% of the total hexamers in the lattice, which is remarkably similar to the 34+/-4% we calculated from the cryoET data. We further quantified the size distributions of these defect regions, showing that they are primarily small defects with a small number of larger areas, which is consistent in both simulation and the cryoET (see new Figure 2-Figure supp). Lastly, we counted the number of free dimeric binding sites and free hexameric binding sites at the outer edge of the lattice. We found more free hexameric binding sites than free dimeric binding sites, which is consistent with experimental analyses from Tan et al., PNAS 2021. However, we note that the experiments saw essentially no free dimer sites, whereas our simulations do have a sizeable portion of free dimer sites. This is because during our assembly simulations, we set the dimer and hexamer binding rates to the same value, such that the dimer is not more rapidly or stably formed relative to the hexamer. To eliminate the free dimer sites, we would need to assemble under more physiologic-like conditions, where the dimer is clearly more stable and faster to form than the hexamer contacts.

We expanded our summary of these results significantly, adding new simulation results showing that the number of defects in simulated lattices is reduced when assembly includes reversible binding that can anneal out some of these defects. We speculate that the relatively high number of incomplete hexamers (high relative to the number required by Euler’s theorem) indicates that the biological assembly process does occur with limited annealing and remodeling. Otherwise, the lattice would have fewer defects. (See revisions on pages 9-10, 10-11 of the manuscript).

Finally, we discuss on page 27 how these defects in the lattice can influence the mechanical stability of the lattice, with new literature references.

3) To confirm that the authors' coarse model is representative of immature CA and not mature CA.

We have performed additional analysis to contrast our coarse model of the immature Gag lattice with the mature Gag capsid. In the new Figure 1-Figure supp 4 we show the immature experimental lattice structure from 5L93.pdb with our overlaid coarse monomer model with interfaces located as they are derived from the protein coordinates. Interfaces are placed based on the average over where residues from each domain are <3.5A from the other partner. The experimental atomic model of the Gag CA-SP1 domains (Gag residues 148371) was extracted from assembled immature HIV-1 particles (Schur et al., Science V353 2016). In the same new figure, we contrast the Gag model for the mature lattice structure as defined in 3J34.pdb (Gag residues 133-363). This experimental atomic model was extracted from cryoEM of the tubular HIV-1 mature capsid assembly. Due to changes in the orientation of the Gag monomers relative to one another in the immature vs mature lattice, our coarse model places interfaces at distinct locations. These models demonstrate a clear difference in geometry due to the experimental structure variations, confirming that our coarse model accurately represents immature CA rather than mature CA. (See revisions on pages 8 and 46 of the manuscript).

4) The argument is made that HIV-1 'utilizes' defects in its lattice to accommodate the protease, providing access to cleavage sites. The authors state that complete, 100% coverage would occlude the cleavage site and prevent proteolysis. Add some discussion about how HIV-2 fits their suggested paradigm.

We appreciate the suggestion to contrast our results on the HIV-1 lattice with the HIV-2 lattice. As we now discuss in the Discussion section, HIV-2 does not have the large vacancy on its surface that HIV-1 has, thus generating higher membrane coverage of 76% ± 8% (Talledge et al., HIV-2 Immature Particle Morphology Provides Insights into Gag Lattice Stability and Virus Maturation, bioRxiv 2022). So while there are still defects and thus room for remodeling, our model would predict significantly slower timescales to dimerization because the edge region for un/rebinding events is significantly diminished. We suggest that the kinetics of Gag interactions should be faster to facilitate remodeling still over a minutes timescale. It would be insightful to directly quantify the dynamics in HIV-2 in future work, and the overall implications for protease dimerization and activation. (See new paragraph on pages 28-29 of the manuscript).

5) Results F, the MFPT formula is an empirical fit. The authors should qualify this by an appropriate measure of the quality of fit, such as an R^2^ measure adjusted for small sample sizes, for example.

Yes, we have now evaluated the Adjusted R-squared value for applying our empirical function to fit our data. R^2^ = 0.98. (See revisions on Figure 4 legend, page 16 of the manuscript).

6) Re-wording the current title.

Taking the reviewer comments and feedback into account, we have now revised the language (primarily removing the term ‘Defects’) of the title to: "Structure of the HIV immature lattice allows for essential lattice remodeling within budded virions".

Reviewer #1 (Recommendations for the authors):The authors found that even in the incompleteness it is extremely unlikely that a lattice will be assembled without a pair Gag-Pols already adjacent (Figure 2B), an observation that clearly suggests that there must exist a mechanism of inhibition until the virion budding since, as stated by the authors, any activation preceding budding is known to reduce infectivity. So, wouldn't it be more interesting to focus on the possible role of the incompleteness of the lattice as a possible mechanism of Pol inhibition? Would it be possible to determine the number of Gag-Pol molecules that would stochastically be located adjacent but with lower-energy interactions which would favor their detachment and separation? How would this be affected by different levels of incompleteness?

We do see that the Gag-Pol pairs occur even when their interaction is completely unfavorable compared to other Gag-Gag interactions (Figure 2-black squares). However, during assembly, all the binding interactions were irreversible, so we performed additional simulations to determine if reversible binding would eliminate these pairs. Now we do see that these unfavorable pairs are largely eliminated, as unbinding can effectively correct for these unstable contacts (Figure 2-red circles). Ultimately, these results imply that suppressing these early dimerization events during assembly does require that the Gag-Pol to Gag-Pol interaction is either highly unfavorable, or if not, the enzymatic activity of the dimer is inhibited in some other way. As the fraction coverage or completeness of the surface increases, the number of pairs does increase due to the higher concentration (See revisions on the page 9 of the manuscript, Figure 2-Figure supp 2).

Would a more-dense assembly lead to Gag-Pol molecules being "trapped" without the possibility of avoiding activation? How would this different point of view be put in context with observations made by Tan A. et al. 2021?

Our simulations show that a denser assembly would indeed lead to a higher number of adjacent Gag-Pol pairs, increasing the likelihood of premature activation. As we discuss now further in regard to HIV-2, a higher membrane coverage would also slow the remodeling dynamics, as we see most of the un/rebinding events happen along the long edge of the vacancy within the lattice. Tan A. et al., 2021 focused on the implications of the edge molecules as being targets for cleavage, rather than its implications for promoting dimerization of the Gag-Pol molecules, which is what we focus on here. In both cases, a denser assembly would lower the size of the edge and thus slow dimerization events and reduce accessibility for cleavage. See new and revised text on page 28.

The title seems a bit disconnected from the main body of the manuscript.

We have revised the title of the manuscript following the suggestions by the reviewers.

Reviewer #2 (Recommendations for the authors):General: Can the authors elaborate on their implicit lipid membrane model and how PIP2 interacts with their protein model? The HIV liposome has an unusual composition, enriched in Cholesterol and Sphingomyelin, and its composition is asymmetric across leaflets. Further, lipid microdomains ostensibly form on the surface and even direct the localization of gag on the membrane surface. MA binds PIP2 which is a part of the maturation cascade, which the authors include as a "stabilization". Does the implicit model treat any of these points directly, and if so how? Where are PIP2 molecules located on the membrane surface? A supplemental figure may help clarify the latter point.

We have hopefully clarified the implicit lipid model approach with additional text and by running new simulations to reproduce our results with an explicit lipid model. This comparison is in Figure 1-figure supplement 3. We still see the same kinetics of assembly, and the proteins still stay bound to the 2D surface throughout the simulation.

On page 30 we better describe our membrane model. First, the membrane is a continuum surface. The only lipids modeled are PI(4,5)P_2_, because they act as binding sites for the Gag proteins. In the explicit lipid model, each lipid is a binding site (i.e. a PI(4,5)P_2_) that diffuses on the surface, and each protein has a lipid binding site that can reversibly attach to these lipids. They are well-mixed on the surface, so we do not assume any lipid rafts/microdomains, and we otherwise do not account for the composition of the membrane. However, because we assume each Gag has bound to a PI(4,5)P_2_, the density is higher in the virion than it would be on the plasma membrane, (assuming the plasma membrane has a density of ~1.5% free PIP2, the density in our simulations is ~3x higher). In the implicit lipid model, we replace explicit diffusing sites with a density field, and the density changes with time as proteins bind/unbind and occupy/free the lipid sites. This method for modeling lipid binding sites is much more efficient.

General: The claim is made that the packaging defects of the immature lattice are similar to those seen experimentally from cryoET. Is there a way to quantify this? On visual inspection alone, they don't seem similar to me. Natural lattice defects seem to be uniformly distributed along the capsid surface, whereas those stemming from this model appear to be a consequence of surface area (i.e., one large continuous region and a corresponding bare region). Some quantification or comparison against published data is necessary.

We have now quantified the structure of our lattice with direct comparison to the cryoET datasets, which were kindly provided by Prof J. Briggs. In both our simulations and the cryoET data, there is a single large continent and a single large gap on the surface, with additional defects present in that large continent.

New analysis is provided in Figure 2-Figure supp, showing that the fraction of incomplete hexamers relative to total hexamers in the lattice is remarkably similar (~35%). The distribution of regions containing incomplete hexamers within the lattice are also quite similar as shown in the Figure supplement. We do note in the supplemental figure legend that the experimental structures have more variability in surface coverage, with some virions having ~300 hexamers vs 530 hexamers, whereas all of our lattices have the same coverage and thus highly similar (550-565) numbers of hexamers. Additionally, we counted the number of free dimeric and hexameric binding sites at the outer edge of the lattice. Our analysis shows that there are more free hexameric binding sites than free dimeric binding sites, which is consistent with experimental findings, although we also see free dimer sites.

(Please see revisions on pages 9-10, 10-11, and 46 of the manuscript)

General: Immature CA has two dimerization sites, one in the N-terminal domain (helix 1) and another in the C-terminal domain (helix 9). Which dimerization site is treated with the virtual particle, purple, shown in Figure 1 panel B.

We determine our interaction sites based on the average position of all the residues in one Gag that interact with another Gag. A residue is considered interacting if it has atoms within 3.5A of an atom on a neighboring Gag. Hence our single site is effectively positioned as an average over any two distinct regions that form interfaces. In our model, the dimerization site is located closer to helix9. We expanded on this in the new Figure 1 figure supplement 4 (page 46).

General – related to the above point: Immature and mature CA is related by differing relative orientations of the N- and C-terminal domains, and this further modulates intermolecular interactions between CA monomers, i.e., different dimerization, trimerization, etc. How do the authors confirm that their coarse model is representative of immature CA and not mature CA? Are there enough details in the model to delineate this feature?

We have added a new figure (Figure 1-Figure supp 4) to compare the coarse models of the immature monomer and mature monomer, which were determined using the same approach. We essentially confirm which representation because of the experimental structure we used; in our paper we used 5L93.pdb, which is an atomic structure derived from immature HIV lattices (Schur et al., Nature 2015). The locations of interfaces in our coarse model represent averages over all the residues in the Gag domains that mediate contact between a Gag pair that form a dimer, and the same procedure for a (different) Gag pair that are in the hexamer contact. We added the analysis of the mature experimental lattice structure from 3J34.pdb. The models display differences in interface geometry due to variations in the Gag positions within the experimental structures.

(Please see the revised text on pages 7 and 46 of the manuscript)

General: Can the authors explain the source of curvature in their model? Does it come from the virtual dimerization site (purple bead in Figure 1 panel B)? Or is it directed by the implicit membrane?

The curvature of our model arises from the binding orientations between all Gag molecules as derived from the experimental atomic structures, we have clarified this in the Figure 1 legend. Even when assembled in solution, our monomers assemble a curved spherical lattice—it is not controlled by the membrane.

General: What are the exact differences between Gag and Gag-pol in the simulations? It is clear from the text that there are parametric differences, but structurally this is not clear.

The only difference between the Gag and Gag-Pol, aside from the label, is during simulations to assemble the initial structures. During the remodeling simulations to calculate the timescales of dimerization (i.e. most of the paper), they are identical, both in terms of structure and kinetics/energetics.

During the initial assembly process for generating the starting structures, we ran a set of simulations where they are identical. We also ran a set of simulations where interactions between Gag-Pol and Gag-Pol are turned off.

We have added text to make this more clear in the Figure 2 legend (page 9), and on page 11 of the results.

General: The cryoET model referenced in the paper does not include the MA domain. How did the authors position their MA bead, and by what criteria was this accomplished, aside from the condition of being normal to the membrane surface? For instance, how was the distance of the bead from the CoM chosen?

The distance between the MA bead and the CoM is set to 2nm, and the vector connecting the bead and the CoM is normal to the membrane surface. This distance was chosen simply to approximately displace the other domains from the membrane surface, and does not impact the structure or kinetics of the remodeling. We revised the text on page 7.

Lines 489-491: The authors suppress multiple nucleation events in the present work, and do not provide any rationale for this choice. It is the opinion of this reviewer that the authors should extend their calculations to include multiple nucleation events. Is there biological evidence that only a single nucleation event initiates spherical particle formation? This point is also alluded to on line 307 as a challenging aspect of this modeling endeavor. If it's real, why not allow it in your models?

Because our goal in this work was to characterize the dimerization of Gag-Pol within an assembled lattice, we wanted to generate initial immature lattice structures that agreed with the known cryoET data, rather than systematically study the process of assembling the Gag lattice. In another study (on bioRxiv doi:10.1101/2023.02.08.527704 and under review), we studied the assembly process, which informed how we could generate the structures we used here. We have added additional text to clarify this point on page 33-34.

We also performed an additional set of assembly simulations here, to show how reversible binding during assembly could reduce the number of incomplete hexamers or defects in the lattice. These results are now included in Figure 2 (page 9) and the results text on page 10-11. We had also conducted simulations (with trace lengths of ~10s) that involved initial lattices composed of two fragments, and compared their remodeling dynamics with those of a single-fragment structure. The simulations showed that the presence of two fragments within the starting structure resulted in slightly slower remodeling dynamics and a longer mean first passage time for Gag-Pol and Gag-Pol dimerization events compared to a single fragment scenario of comparable length. We note this in the Discussion on page 28.

Lines 530-539: The argument is made that HIV-1 'utilizes' defects in its lattice to accommodate the protease, providing access to cleavage sites. The authors state that complete, 100% coverage would occlude the cleavage site and prevent proteolysis. This begs the question about HIV-2, or other retroviruses, which is known to have a significantly more complete immature lattice. Can the authors add some discussion about how HIV-2 fits their suggested paradigm? This would be an interesting inclusion for readers.

We thank the reviewer for bringing this interesting point to our attention. We have now added a paragraph in the Discussion section to discuss how we expect dimerization to be significantly slower in the HIV-2 lattice, albeit still possible given that the surface coverage is still ~76%.

(Please see the revision on pages 28 of the manuscript)

Figure 4 panel C – According to the caption, the fits in this panel were taken from data points in which 75% of the trajectories produced dimerization events, which yields a linear fit of the data. Is it valid to take only those data points where 75% of the trajectories produced dimerization? Since if the additional data points are included, the trend changes completely and is non-linear. Some additional discussion about this choice would be helpful – in the caption or otherwise.

We have added further text to clarify this decision. We chose the 75% cut-off, because the points that had 75% completed events or more still agreed well with the power-law trends in the functional fits that we get if we fit the statistically most reliable points, i.e. those with 100% of events completed. We know that reporting on a mean value when the distribution is incomplete and truncated by an upper bound (for us~23 seconds) would, by construction, result in underestimates of the timescales, as the remaining fraction of the distribution will necessarily slow the mean. Hence we chose a cut-off that preserved the agreement that we trust from the 100% completed points, and our physically motivated, albeit empirical functional fit provides a quantitative model for extrapolation. We revised the text on page 16 accordingly.

General: there is a lot of jargon, which makes the paper difficult to decipher for a general audience. Can the authors expand and/or elaborate where necessary, e.g., iPALM.

We have tried to eliminate jargon and make the text more accessible to a general audience.

– ‘reaction-diffusion’ is replaced by either computer simulations or spatio-temporal models, and then given more context and description in the introduction.

– iPALM is replaced by ‘time-resolved microscopy’ or ‘super-resolution imaging’, and only described in more detail in the Methods, where we provided more context for its role in calculating an auto-correlation function.

– Defined ODE as ordinary differential equation – Explained the HALO/SNAP tag

– Defining intrinsic rate relative to a macroscopic, or bulk biochemical rate on page 20.

– Explained Brownian updates on page 30.

Reviewer #3 (Recommendations for the authors):My main comments are summarized in the "public review". Privately, and despite the evident seriousness of this study, I found the results a little underwhelming. That is, there does not seem to be anything particularly surprising, or conceptually new. This, therefore, raises questions of significance, and how the collective impact of the results will serve to drive the HIV field forward, either by comprehensively solving an outstanding problem or by opening up a new avenue of research. Of course, there may be some context here that I have missed, but at present, I am struggling to see it.

Since the formation of a dimer of proteases between Gag-Pol polyproteins is required for the infectivity of the virus, we think that our work establishes an essential foundation for the physical mechanisms of this dimerization process. We have comprehensively shown that the dimerization process can occur at physiologically relevant timescales even if the protease domains are locked into the immature lattice. The only other alternative that we see is that the protease dimer forms prior to budding but is actively inhibited. We further show here that dimers are statistically likely to form given the abundance of Gag-Pol relative to Gag, and thus would need to be suppressed in some way to prevent early activation. Our work thus indicates that new experiments would be needed to determine this (presumably) molecular mechanism.

Importantly, by comparing our simulations to experiments, we establish bounds on the stability (free energy) of the Gag-Gag contacts in the lattice and the Gag binding kinetics, which has otherwise been inaccessible from either experiment or alternative modeling approaches. Thus, any further modeling studies (including our own) should be constrained by these quantitative bounds when studying steps in the formation and remodeling of the Gag immature lattice.